# A synthetic microbial biosensor for high-throughput screening of lactam biocatalysts

Soo-Jin Yeom[1], Moonjeong Kim[1,2], Kil Koang Kwon[1], Yaoyao Fu[1], Eugene Rha[1], Sung-Hyun Park[1,3], Hyewon Lee[1], Haseong Kim[1,3], Dae-Hee Lee [1,3], Dong-Myung Kim[2] & Seung-Goo Lee[1,3]

Biocatalytic cyclization is highly desirable for efficient synthesis of biologically derived chemical substances, such as the commodity chemicals ε-caprolactam and δ-valerolactam. To identify biocatalysts in lactam biosynthesis, we develop a caprolactam-detecting genetic enzyme screening system (CL-GESS). The *Alcaligenes faecalis* regulatory protein NitR is adopted for the highly specific detection of lactam compounds against lactam biosynthetic intermediates. We further systematically optimize the genetic components of the CL-GESS to enhance sensitivity, achieving 10-fold improvement. Using this highly sensitive GESS, we screen marine metagenomes and find an enzyme that cyclizes ω-amino fatty acids to lactam. Moreover, we determine the X-ray crystal structure and catalytic residues based on mutational analysis of the cyclase. The cyclase is also used as a helper enzyme to sense intracellular ω-amino fatty acids. We expect this simple and accurate biosensor to have wide-ranging applications in rapid screening of new lactam-synthesizing enzymes and metabolic engineering for lactam bio-production.

[1] Synthetic Biology and Bioengineering Research Center, KRIBB, Daejeon 34141, Republic of Korea. [2] Department of Chemical Engineering and Applied Chemistry, Chungnam National University, Daejeon 34134, Republic of Korea. [3] Department of Biosystems and Bioengineering, KRIBB School of Biotechnology, University of Science and Technology, Daejeon 34113, Republic of Korea. Correspondence and requests for materials should be addressed to S.-G.L. (email: sglee@kribb.re.kr)

dentifying biocatalysts, in particular, enzymes or pathways that play key roles in biosynthesis of non-native molecules of interest, is important to the industrial synthesis of chemical products. The discovery and engineering of enzymes or pathways involved in the synthesis of a desired product are often limited by a lack of sufficiently sensitive and rapid screening tools for the identification of candidate genes from large natural or synthetic gene libraries[1–3]. Genetically encoded biosensors have untapped potential as tools for screening enzymes and pathways; thus, extensive efforts have been made to develop high-throughput screening (HTS) biosensors equipped with fluorescence-based genetic circuit devices. A key component in such devices is the ligand-inducible transcription factor (TF). In nature, a wide variety of TFs can specifically recognize small molecules and alter gene transcription at their targeted promoters. Numerous TF-based biosensors with ligand specificity and dynamic detection ranges are already available for sensing various small molecules[4]. Such biosensors are straightforward and powerful tools for detecting target molecules or their intermediates in investigations of enzymes and biosynthesis pathways[5–9]. Recently, many efforts using TF-based sensors in HTS have focused on altering TF specificities or sensitivities[1]. For instance, AraC of *Escherichia coli*, which originally senses arabinose, was engineered to detect mevalonate—a key intermediate of isoprenoid pathways—which enabled the screening of cells with increased mevalonate synthesis[10].

Lactams are industrially important chemicals that are used in polyamide production[11,12]. δ-Valerolactam and ε-caprolactam (CL) are converted from ω-amino fatty acids (5-amino-valeric acid (5-AVA) or 6-aminocaproic acid (6-ACA)) and are used as precursors for the production of nylon-5, nylon-6, and nylon-6,5, which are used to manufacture tire cords, carpeting, plastics, and food-packaging materials[13]. ε-Caprolactam is most widely used to produce nylon-6 and is mainly produced through Beckmann rearrangement of the cyclohexanone oxime in the presence of fuming sulfuric acid, at 90–120 °C. Biorenewable routes towards ε-caprolactam from fermentation-derived lysine, muconic acid, adipic acid, and 6-ACA have been discussed[14]. Additionally, the production of 6-ACA by direct fermentation of glucose has also been reported[15].

Although several ω-amino fatty acids are biologically produced through biosynthetic pathways, complete biosynthetic pathways capable of synthesizing lactams are mostly unknown owing to the lack of enzymes able to catalyze the last ring-cyclization step. To date, only two enzymes that can be used for lactam biosynthesis have been reported to carry out this step: *Candida antarctica* lipase B (CALB)[16] and *Streptomyces aizunensis* acyl-CoA ligase (ACL)[10]. However, CALB requires an anhydrous condition, high temperature, and long reaction time; thus, it is not suitable for lactam biosynthesis. ACL exhibits a broad substrate spectrum and has been used for cyclizing 4-aminobutyric acid, 5-AVA, and 6-ACA into γ-butyrolactam, δ-valerolactam, and ε-caprolactam, respectively[11]. Both 6-ACA and its cyclized form, ε-caprolactam, are non-natural compounds; a previous report designed and proposed two biosynthetic pathways for fermentative production of 6-ACA[15]. However, there is a lack of efficient and specific enzymes capable of producing ε-caprolactam from 6-ACA through enzymatic conversion and microbial fermentation.

In this study, we aim to investigate the lactam biosynthesis pathway in greater detail through the identification of certain enzymes involved, using HTS biosensors. To this end, we firstly designed and engineered a lactam-detecting biosensor, termed caprolactam-detectable genetic enzyme-screening system (CL-GESS), and then carried out HTS of ε-caprolactam-converting cyclases from diverse metagenomes. To improve the signal-to-noise ratio and sensing sensitivity, a transcriptional regulator,

NitR, is engineered and used in the CL-GESS to identify a cyclase for ε-caprolactam or valerolactam biosynthesis from 6-ACA or 5-AVA, respectively. Finally, we determined the X-ray crystal structure of the newly identified cyclase to provide insight into its cyclization activities. Using the cyclase, we developed a genetically encoded biosensor to sense ω-amino fatty acids (5-ACA or 6-AVA). We presented a cyclase that converts 6-ACA to ε-caprolactam, which will open opportunities for the development of a bioprocess to produce lactams and nylons.

## Results

**Design and construction of a lactam biosensor**. The experimental strategy used to develop the CL-GESS is illustrated in Fig. 1. Our strategy included (a) engineering of a transcriptional regulator by fluorescence-activated cell sorting (FACS), (b) combinatorial analysis and optimization of a promoter or ribosomal-binding site (RBS) for controlling a regulator, (c) searching for a binding site on the regulator, and (d) optimization of a reporter gene. To develop a biosensor that responds to ε-caprolactam, we selected lactam-responsive TFs in nature and integrated them into a genetic circuit. In the actinomycete *Rhodococcus rhodochrous* J1[17–19] which is used in the industrial production of acrylamide and nicotinamide, nitrilase is strongly induced in the presence of ε-caprolactam[19]. In *R. rhodochrous*, nitrilase is encoded by the *nitA* gene in the nitrile degradation operon, which is positively regulated by NitR, which is in turn activated by isovaleronitrile or ε-caprolactam[17]. As *Alcaligenes faecalis* JM3 also contains a nitrile degradation operon, we examined the applicability of the *A. faecalis nitR* regulatory subunit (nitrile degradation operon) in the development of the CL-GESS by assessing the induction of *nitA* gene expression in the presence of various effector molecules. NitR responded solely to ε-caprolactam, which activated *nitA* gene expression (Supplementary Fig. 1). Based on this result, we transformed the *nitR* (*E. coli* codon-optimized) regulatory system of *A. faecalis* in the nitrile degradation operon and the *nitA* promoter ($P_{nitA}$) into *E. coli* to develop a synthetic CL-GESS for the specific detection of lactam (Fig. 1).

**Optimization and characterization of the CL-GESS**. We next optimized the CL-GESS to enhance its sensitivity. We constructed five different CL-GESSs in this study (Supplementary Fig. 2). The first, CL-GESSv1, contained the *E. coli* codon-optimized *nitR* gene under the control of a constitutive J23100 promoter in the direction opposite to that of transcription of the putative $P_{nitA(748)}$ promoter–*eGFP* fusion (Supplementary Fig. 2). In a second plasmid, the eGFP reporter gene in CL-GESS (CL-GESSv1) was replaced with superfolder GFP (sfGFP) to produce CL-GESSv2. CL-GESSv2 showed higher fluorescence to all tested concentrations of ε-caprolactam than did CL-GESSv1 (0.5–50 mM, Fig. 2a). We further investigated the ability of the $P_{nitA}$ region in CL-GESS to control a reporter, because we speculated that a strong promoter containing a NitR-binding site would be located close to the *nitA* gene. Initially, we used the 748-bp DNA fragment upstream of *nitA* as the putative $P_{nitA}$ to drive sfGFP expression in the CL-GESS plasmid, because the $P_{nitA}$ region in genomic DNA had not yet been annotated. Additionally, there was no information on the binding of NitR to the $P_{nitA}$ region. One hundred-bp, 200-bp, or 300-bp truncations of the 748-bp $P_{nitA}$ fragment from the RBS (T7RBS in this study) of the reporter gene were studied. Based on reporter experiments of the $P_{nitA}$ promoter fragments, $P_{nitA}$ was found to be located within 200 bp from the RBS as CL-GESSv3 showed strong fluorescence (Supplementary Fig. 3a).

The putative NitR-binding site was palindromically located in the −35 and untranslated regions of $P_{nitA}$ (Supplementary Fig. 3b, c). It is important to note that sequences at positions −35 (TTCATC) and −10 (TACACT) upstream of the transcription start site were similar to the actinomycete (mainly streptomycete) consensus promoter sequence TTGAC(A/G)-17 bp-TAg(A/G)(A/G)T[18]. In addition, a single transcription initiation site was identified 74-bp upstream of the first ATG codon of GFP (Supplementary Fig. 3d). Deletion of NitR from the CL-GESS plasmid resulted in complete loss of $P_{nitA}$-induced green fluorescence (Supplementary Fig. 4), indicating that NitR activates the transcription of the target gene at its targeted promoter $P_{nitA}$.

To further optimize *NitR* expression in CL-GESSv3, which contained sfGFP and truncated $P_{nitA}$, we substituted the promoter and RBS with various synthetic promoters and RBSs of different strengths (rank order of promoter strength: J23100>J23106>J23114; rank order of RBS strength: B0030>B0034, and T7RBS) (http://parts.igem.org/Promoters/Catalog/Anderson, Supplementary Table 1). The plasmids CL-GESS J23114-B0034(CL-GESSv4) showed the highest fold change in fluorescence in the presence of ε-caprolactam (Fig. 2b).

To quantitatively assess the response of CL-GESSv4 to ε-caprolactam, we measured fluorescence at the single-cell level in *E. coli* DH5α cells harboring CL-GESSv4 in Luria-Bertani (LB) medium, at various ε-caprolactam concentrations (up to 30 mM). Fluorescence was observed only in the presence of ε-caprolactam

(Fig. 2c). The minimal concentration of ε-caprolactam required to activate NitR was 500 μM; moreover, a tight correlation was observed between fluorescence intensity and ε-caprolactam concentration. To examine the ligand specificity of CL-GESSv4, we applied it to various substrate and precursor molecules involved in lactam biosynthesis. CL-GESSv4 sensed ε-caprolactam, ε-caprolactone, cyclohexanone, *N*-acetylcaprolactam, δ-valerolactam, δ-valerolactone, benzonitrile, and isovaleronitrile, whereas detected none of the intermediates in the proposed ε-caprolactam biosynthesis pathway (Fig. 2d, Supplementary Fig. 7a). Even at high concentrations (30 mM), intermediates, such as L-lysine, 5-AVA, and 6-ACA did not induce sfGFP expression above the background level. To investigate the relationship between the intracellular ε-caprolactam level and sensor output, we measured intracellular ε-caprolactam. Intracellular ε-caprolactam was 0.035–0.27 μmole/mg wet cells when we added 1–30 mM caprolactam extracellularly (Supplementary Table 2). As the concentration of external ε-caprolactam increased, the intracellular ε-caprolactam concentration also increased (Supplementary Table 2). When we washed the cells twice or three times with 50 ml of saline, intracellular ε-caprolactam was not detected. These observations supported that the uptake and release of ε-caprolactam appear to be equilibrated by passive intracellular transport.

In addition, we evaluated the applicability of CL-GESSv4 to various Gram-negative bacteria, including *Pseudomonas putida* strains KT2440 and S12, and *Ralstonia eutropha*. The backbone

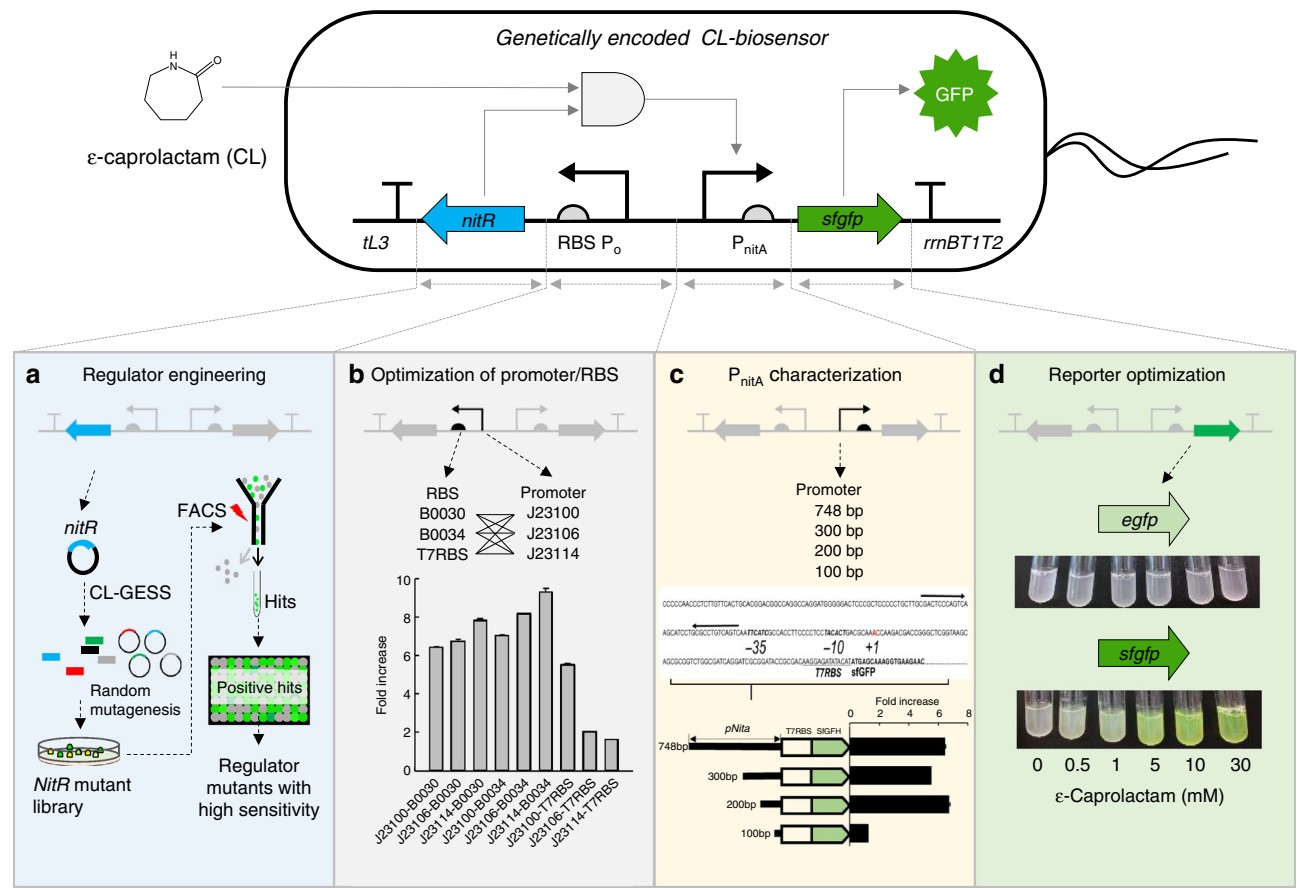

**Fig. 1** Schematic representation of strategy for development of CL-GESS. ε-Caprolactam activates NitR derived from CL-GESS, which then activates the $P_{nitA}$ promoter and *sfgfp* gene expression in CL-GESS, resulting in fluorescence emission. **a** Schematic illustration of the screening procedure for engineering the NitR regulator by random mutagenesis. **b** Systematic combinatorial analysis and optimization of promoter and RBS for *NitR* expression. **c** Identification of the $P_{nitA}$ region by gene truncation. **d** Reporter change in CL-GESS

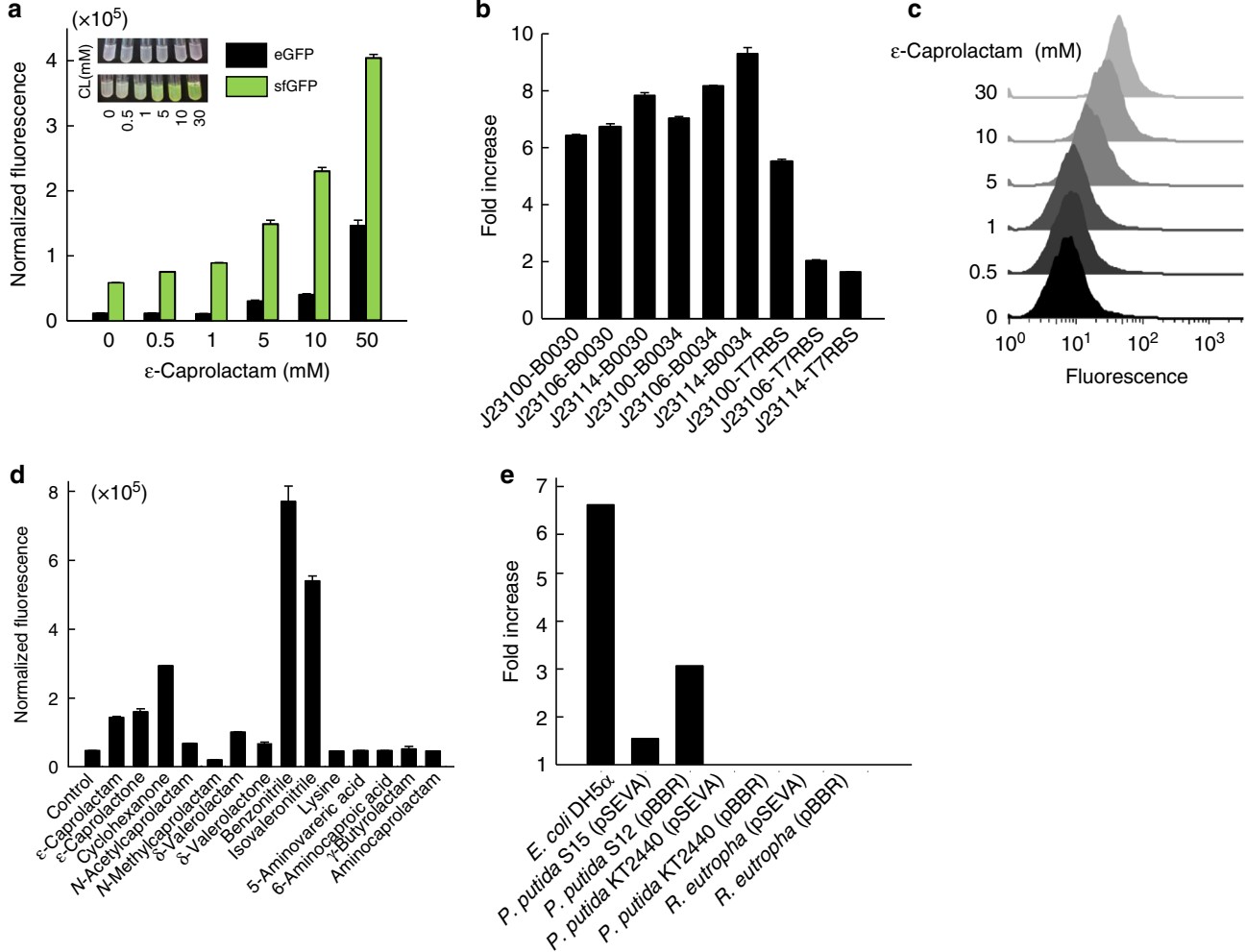

**Fig. 2** Quantitative analysis of CL-GESS and CL-GESS variant responses. **a** Effect of reporter protein replacement in CL-GESS. **b** Promoter and RBS study of *NitR* expression in CL-GESS. Systematic combinatorial analysis and optimization of *NitR* expression was carried out using constitutive BioBrick promoters with known relative strength (http://parts.igem.org/Promoters/Catalog/Ecoli/Constitutive, Supplementary Fig. 2). Values represent means ± SDs of three independent experiments. **c** Correlation between concentration of exogenous ε-caprolactam and specific fluorescence measured 16 h after addition of ε-caprolactam at the indicated concentrations to *E. coli* cells harboring CL-GESS. Representative histograms from three independent rounds of FACS are shown. **d** Ligand specificity of CL-GESS. Various lactam or lactone compounds were detectable at a concentration of 1 mM by CL-GESS. Values represent means ± SDs of three independent experiments. **e** Fluorescence signal of CL-GESS in various hosts. Values represent means ± SDs of three independent experiments

vector of CL-GESSv4 was replaced with pBBRBB[20] or pSEVA (http://seva.cnb.csic.es) containing RK2 or pBBR1 *ori*, respectively. *P. putida* S12 containing pBBRBB-based CL-GESSv4 showed a three-fold increase in signal in the presence of ε-caprolactam, whereas S12 transformed with pSEVA-CL-GESSv4 exhibited a 1.5-fold increase (Fig. 2e). *P. putida* KT2440 and *R. eutropha* harboring pBBRBB-based or pSEVA-based CL-GESS showed no fluorescence response to ε-caprolactam (Fig. 2e). These result shows the possibility that CL-GESS system can be applied in *P. putida* with various advantages for natural product biosynthesis, such as a versatile intrinsic metabolism with diverse enzymatic capacities, and outstanding tolerance to xenobiotics[21]. For these reasons, the genetic circuit-based sensor in *P. putida* should be further studied.

**Engineering the NitR TF.** To improve the sensitivity of CL-GESSv4 further, we engineered the NitR transcriptional regulator. To this end, we designed a single cell-based screening strategy for a library of NitR with random mutations. We selected hits with a high fluorescence signal in the presence of ε-caprolactam and

removed false positives that exhibited fluorescence in the absence of ε-caprolactam. After three rounds of screening, the highest 0.5% of phenotypes were selected as putative hits and their fluorescence signals were verified against ε-caprolactam in liquid LB medium (Supplementary Fig. 5a). Among them, a NitR double mutant, CL-GESS$_{NitR-L117F/P133S}$, showed increased fluorescence signal when exposed to ε-caprolactam at a low concentration of 50 μM as minimum while wild-type CL-GESS showed increased fluorescence signal more than 500 μM (Fig. 3a).

We further engineered the double mutant (CL-GESS$_{NitR-L117F/P133S}$) to maximize the sensor sensitivity. To find a mutation to generate NitR mutants more sensitive to ε-caprolactam, we firstly tested the P133 location. However, when we changed the P133 to serine, there was no change in sensitivity of CL-GESS to ε-caprolactam, but the background fluorescence was increased. The single-point mutant CL-GESS$_{NitR-L117F}$ exhibited strong fluorescence in the presence of ε-caprolactam (Fig. 3b). Replacement of L117 in the NitR regulator with the non-polar aliphatic amino acid alanine or a charged polar amino acid, such as arginine or glutamate, abolished the ability of the CL-GESS to detect

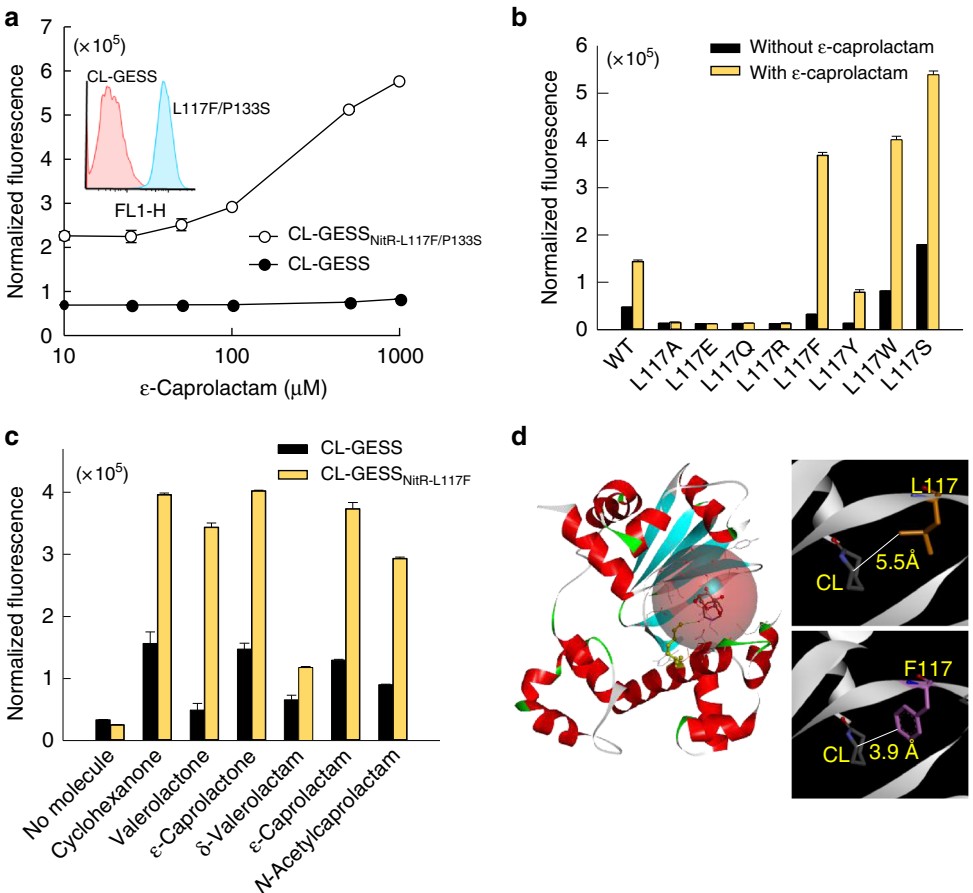

**Fig. 3** Engineering of the NitR regulator. **a** Quantitative analysis of the response of CL-GESS and CL-GESS double mutant (S133P, L117F) to ε-caprolactam. **b** Quantitative analysis of the response of CL-GESS and CL-GESS$_{L117}$ variants to ε-caprolactam. **c** Ligand specificity of mutant CL-GESS$_{L117F}$. **d** Homology modeling of CL-GESS and CL-GESS$_{L117F}$. Values represent means ± SDs of three independent experiments

ε-caprolactam (Fig. 3b). Interestingly, the substitution of L117 with aromatic residues, as in CL-GESS$_{NitR-L117F}$, CL-GESS$_{NitR-L117Y}$, and CL-GESS$_{NitR-L117W}$, induced a marked increase in the fluorescence signal of 11.4-fold, 5.6-fold, and 4.9-fold, respectively, in the presence of ε-caprolactam as compared to the 3-fold increase observed for wild-type CL-GESS (Fig. 3b). CL-GESS$_{NitR-L117F}$ also showed enhanced fluorescence for cyclohexanone, δ-valerolactone, ε-caprolactone, δ-valerolactam, and N-acetylcaprolactam (Fig. 3c). CL-GESS$_{NitR-L117F}$, but not wild-type CL-GESS, was able to detect δ-valerolactone, δ-valerolactam, and n-acetylcaprolactam (Fig. 3c). CL-GESS$_{NitR-L117}$ variants (L117F, L117Y, and L117W) also yielded strong signals in the presence of various lactones or lactams rather than nitrile compounds (Supplementary Fig. 6). These results suggested that bulky aromatic residues in position 117 may interact with lactam or lactone substrates. To determine the amino acid residues of NitR that are important for the interaction with ε-caprolactam, a homology model of *A. faecalis* NitR was constructed based on the crystal structure of *Vibrio cholerae* O395 ToxT (Protein Data Bank (PBD) entry 3GBG)[22] as the closest sequence among the known structures of AraC-type regulators. Although the sequence identity between NitR and ToxT was relatively low (18.1% identity, 38.2% similarity), the sequence identity (23% identity, 43% similarity) of the N-terminal domain (substrate-binding site) between two regulators can result in useful homology model. Molecular docking was attempted to estimate the substrate interactions in the *A. faecalis* NitR active site (Fig. 3d). Amino acid L117 of NitR was estimated to be located on one side of the

active pocket (Fig. 3d) and the mutations to aromatic amino acids was estimated to result in a shorter distance between a docked ε-caprolactam and the aromatic side chains in mutant NitR (Fig. 3d). These results suggest that an aromatic amino acid at position 117 closely interacts with lactam or lactone substrates in the active pocket. Although the L117W mutant with larger side chains appeared more sensitive to ε-caprolactam (Supplementary Fig. 6), it also showed a higher background signal. Therefore, we used L117F mutant for the screening of ε-caprolactam producing enzymes because high background of L117W mutant could be a problem during the screening of large library.

**HTS of a metagenome library using CL-GESS$_{NitR-L117F}$.** A metagenome is a potentially substantial reservoir of valuable enzymes or biocatalysts that could be targeted by CL-GESS$_{NitR-L117F}$. Various intermediates in the ε-caprolactam synthesis pathway, including L-lysine, 5-AVA, and ACA did not induce sfGFP expression of CL-GESS$_{NitR-L117F}$ in solid or liquid LB medium, suggesting that the highly sensitive CL-GESS$_{NitR-L117F}$ can be used for HTS of new enzymes in the ε-caprolactam biosynthesis pathway (Supplementary Fig. 7). We carried out CL-GESS-based HTS of a metagenomic fosmid library constructed using tidal flat sediments. To this end, we firstly transformed a plasmid encoding CL-GESS$_{NitR-L117F}$ into *E. coli* cells to screen a cyclase that could convert 6-ACA into ε-caprolactam. Cells grown in LB medium containing 10 mM 6-ACA were sorted by flow cytometry. After removing false positives by culturing recovered cells in fresh LB medium without 6-ACA, a total of 10$^6$ cells were

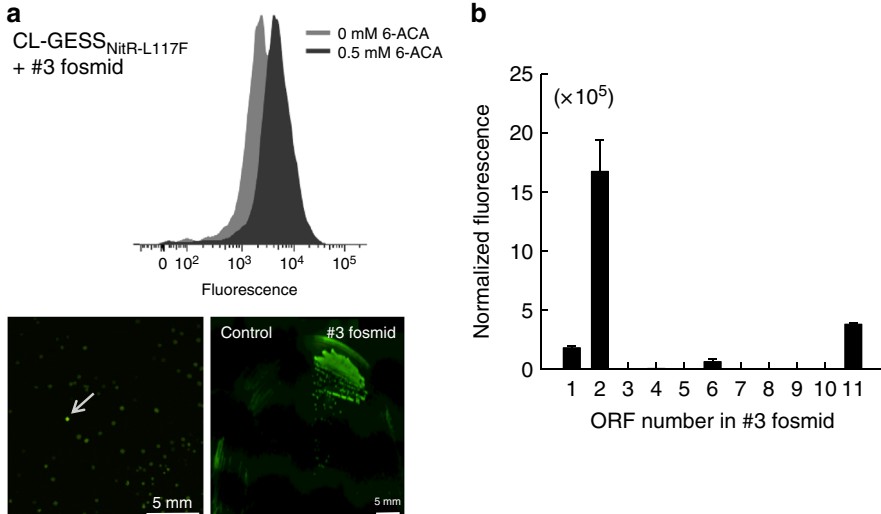

**Fig. 4** HTS of cyclases from metagenome library using CL-GESS_L117F. **a** Selection of a flow cytometry histogram from metagenome library containing 500 μM 6-ACA and selection of a strongly fluorescent colony (positive hit) on LB agar containing 10 mM 6-ACA. The scale bar is 5 mm. **b** Analysis of fluorescence in a clone with 11 ORFs in pET28a(+) from a metagenomic fosmid using CL-GESS_L117F to assess ε-caprolactam-converting activity from 10 mM 6-ACA over 24 h. Values represent means ± SDs of three independent experiments

obtained (Supplementary Fig. 5). These cells were cultured in LB medium containing 10 mM 6-ACA and sorted again based on fluorescence intensity, yielding 27 hits showing stronger fluorescence on LB solid plate than the control. The 27 hits were grown on solid LB medium containing 6-ACA, and one colony (named fosmid #3), with the strongest fluorescence, was selected by microscopy (Fig. 4a). Flow-cytometric analysis revealed that fosmid #3 with CL-GESS_NitR-L117F exhibited stronger fluorescence in the presence of 6-ACA than did cells harboring the empty fosmid vector (Fig. 4a, Supplementary Fig. 8). This suggested that fosmid #3 contained at least one open-reading frame (ORF) encoding an enzyme that catalyzed the biosynthesis of ε-caprolactam from 6-ACA. Therefore, we isolated DNA from fosmid #3 cells, and then completely sequenced it using the shotgun method.

The DNA from fosmid #3 contained a 31-kb DNA fragment that showed up to 88% identity to *Citrobacter* genome sequences (Supplementary Table 3). At least one ORF encoding a putative cyclase was identified, supporting the results of the functional screen. Of the 25 ORFs in clone #3, we selected 11 ORFs that were up to 500 bp (Supplementary Table 4) and subcloned them into the pET28a(+) plasmid. *E. coli* cells containing CL-GESS_NitR-L117F were transformed with the pET28a(+) plasmid encoding an individual ORF to detect the fluorescence signal corresponding to the conversion of 6-ACA to ε-caprolactam. We found that an ORF encoding 3-hydroxybutyrate dehydrogenase from *Citrobacter freundii* (CF3HBD) was positive for this activity (number 2 in Fig. 4b).

**Characterization of CF3HBD.** To corroborate its cyclase activity, CF3HBD was expressed and purified as a 27-kDa protein (Supplementary Fig. 10a), and its cyclization activity was calibrated by detection of ε-caprolactam by liquid chromatography–mass spectrometry (LC–MS; Fig. 5a and Supplementary Fig. 9b) and nuclear magnetic resonance (NMR; Fig. 5b). Importantly, this reaction was performed using purified enzyme and in the absence of any cofactors, such as ATP and NADH. Furthermore, the reaction mixture was extracted with ethyl acetate to purify the reaction product from the other materials. LC–MS analysis revealed that the purity of the obtained caprolactam was >99% (Supplementary Fig. 9a). Thus, based on the product confirmed

by NMR, LC–MS, and IR (infrared spectroscopy) analyses, we concluded that the promiscuous enzyme catalyzed the cyclization of 6-ACA in the absence of any cofactor (Supplementary Fig. 9a, c, d). CF3HBD showed 81%, 82%, 81%, 56%, and 28% sequence identity with 3HBD from *Serratia marcescens*, *Klebsiella pneumoniae*, *Enterobacter* sp., *Rhodobacter sphaeroides*, and *Pseudomonas lemoignei*, respectively (Supplementary Fig. 10b). We cloned two additional 3HBDs from *S. marcescens* (SM3HBD) and *Enterobacter* sp. (ES3HBD) and determined their cyclase activity after purification (Supplementary Fig. 10a) using 6-ACA as a substrate; both enzymes showed cyclization activity towards 6-ACA to produce ε-caprolactam as well (Supplementary Fig. 10c). However, the 3HBDs from *R. sphaeroides* and *P. lemoignei*, with low sequence identity, showed no cyclization activities.

Basically, the screened CF3HBDs belong to the NAD(P)H-dependent short-chain dehydrogenase family, and they have cyclization activity as unexpected promiscuous activity. D-3-Hydroxybutyrate dehydrogenase catalyzed the reversible and stereospecific oxidation of D-3-hydroxybutyric acid to acetoacetate using NAD$^+$ as a coenzyme[23]. Thus, we further investigated simple enzyme characteristics in terms of the effects of pH and temperature, metals, and coenzymes on enzyme activities. The maximum dehydrogenation activity of CF3HBD towards 3-hydroxybutyrate was observed at pH 8.0 and at 40 °C (Supplementary Fig. 11a, b). The maximum cyclization activity of CF3HBD towards 3-hydroxybutyric acid was observed at pH 7.5 and 40 °C (Supplementary Fig. 11a, b). The cyclization activity of CF3HBD did not require metals or cofactors in vitro (Supplementary Fig. 12a, b). The time course analyses for the production of ε-caprolactam were conducted using 1 mM 6-ACA and CF3HBD. The enzyme concentrations used were 0.1, 0.25, and 0.5 mg/ml and yielded molar conversions of 17.9%, 34.6%, and 47.3%, respectively (Supplementary Fig. 13). No detectable intermediate was obtained during this reaction. In addition, we found that CF3HBD could catalyze the degradation of the cyclic amide group in ε-caprolactam, generating 6-ACA as a reaction product (Supplementary Fig. 14).

**Crystal structure of the CF3HBD and homology modeling.** To further characterize the cyclization activity, we determined the crystal structure of CF3HBD in complex with NAD$^+$ at a

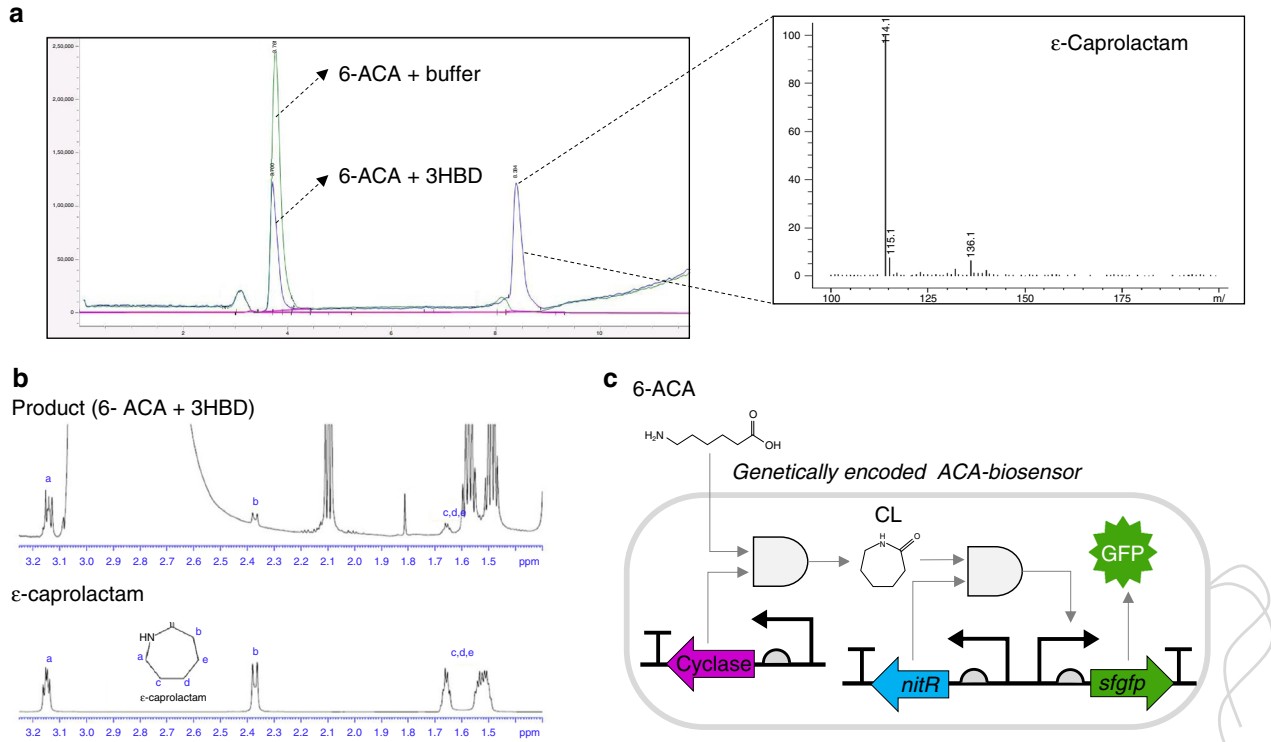

**Fig. 5** Cyclase activity. **a** LC–MS analysis and mass configuration of ε-caprolactam produced by CF3HBD from 6-ACA (blue peak) and 6-ACAwithout CF3HBD as a control (green peak). Analysis of the reaction mixture by LC–MS. The reaction was carried out using 0.5 mg/ml purified CF3HBD and 1 mM 6-ACA as the substrate at 35 °C with 50 mM HEPES buffer (pH 7.5) **b** NMR analysis of the reaction mixture for 3HBD in the presence of 6-ACA. The reaction was carried out using 0.5 mg/ml purified CF3HBD and 1 mM 6-ACA as the substrate at 35 °C with 50 mM HEPES buffer (pH 7.5). **c** Schematic representation of the ACA-GESS system consisting of CL-GESS and cyclase in *E. coli*. Intracellular ε-caprolactam compounds were generated by 3HBD from ω-fatty acids, such as 5-AVA and 6-ACA, and were visualized by sfGFP, whose expression was induced by the ε-caprolactam–NitR complex

resolution of 2.29 Å (Supplementary Fig. 15). We found that the asymmetric unit had a tetramer structure with two $NAD^+$-free subunits (Fig. 6a). However, the crystal structure of CF3HBD in complex with ε-caprolactam or 6-ACA under various conditions could not obtain diffraction data. The crystal structure of the CF3HBD–$NAD^+$ complex was refined to an R factor of 19.1% ($R_{free} = 23.6$%) for all data observed, without any cutoff in the resolution range 50–2.3 Å. A Ramachandran plot of the four polypeptides showed that all of the main-chain atoms fall within the allowed regions, with 96.1% of residues in the most favored regions and 3.6% of residues in additionally allowed regions. The refined structure was in good agreement with the X-ray crystallographic statistics for bond angles, bond lengths, and other geometric parameters (Table 1). Nearly all of the residues were well defined in the electron-density map, except the unidentified electron density located in the pocket near the nicotinamide group of $NAD^+$, which corresponded to the active site of CF3HBD (Fig. 6b). Based on its position, this density could represent a molecule of ε-caprolactam transiently captured by the enzyme during crystal packing. Molecular docking of CF3HBD with 6-ACA and ε-caprolactam in the unidentified electron density was achieved using the holo form of CF3HBD (Fig. 6c, d). The affinities of 6-ACA and ε-caprolactam to CF3HBD were −4.3 and −3.7 kcal/mol, respectively.

**Determination of the active site residues by mutagenesis**. To identify the catalytic residues involved in the dual activities of CF3HBD, we conducted a ligand-docking study of the linear form of 6-ACA with the crystal structure. The docking results suggested that 10 residues (Q91, S139, V140, H141, K149, Y152, Q193, W184, V190, and Q193) located within 4.0 Å of the center

of the docked substrate are active-site residues. These residues were selected as candidate determinant residues for enzyme activity. The selected 10 residues were separately replaced with alanine or glutamate, and the wild-type and all mutant 3HBD were expressed and purified by His tag affinity chromatography as a single band with a molecular mass of ~27 kDa in sodium dodecyl sulfate–polyacrylamide gel electrophoresis (SDS–PAGE) (Supplementary Fig. 16). Alanine substitution of the catalytic residues, such as Q91, S139, and H141 completely abolished catalytic activity towards both 3-hydroxybutyric acid and 6-ACA, which suggested that dehydrogenation activity and cyclization activity share catalytic site (Table 2). Alanine substitution at V140, Y152, and W184 resulted in lack of activity towards 3-hydroxybutyric acid, whereas these mutants showed cyclization activities for 6-ACA of more than 80%. Interestingly, the Y152A mutant showed the highest cyclization activity, with a three-fold increase, for 6-ACA. Thus, we determined the kinetic parameters using the Michaelis–Menten and Lineweaver–Burk plots of the wild-type and Y152A-mutant enzymes toward 6-ACA (Supplementary Fig. 17). The Y152A mutant had about 3.4-fold higher catalytic efficiency than that of wild-type enzyme (Table 3). This result suggested that an alanine residue may be preferred rather than the bulky aromatic amino acid to allow 6-ACA in the active site, but not 3-hydroxybutyric acid.

**CL-GESS as a sensor for ω-amino fatty acid cyclization**. We investigated whether CL-GESS$_{L117F}$ can be used as a biosensor that reflects the amount of ε-caprolactam formed from 6-ACA as a fluorescence signal detectable by flow cytometry (Fig. 5c). We co-transformed *E. coli* EPI(DE3) cells[24] with the CL-GESS$_{L117F}$ sensor and the other plasmid expressing CF3HBD under the

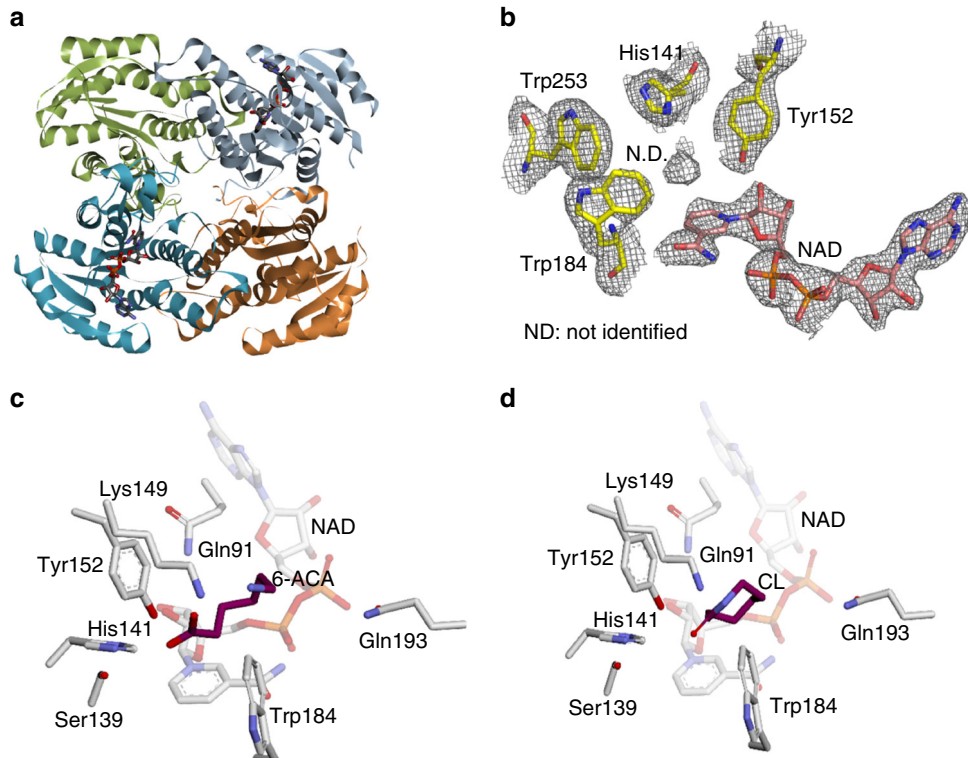

**Fig. 6** Crystal structure and homology model of CF3HBD. **a** Overall structure of CF3HBD. **b** An electron density map for the active site region. Electron densities were contoured using COOT. Unknown molecule observed in the substrate-binding pocket. The 2mFo-DFc electron density map is shown as a gray mesh contoured at $1.0\sigma$. **c** Proposed model for 6-ACA-binding by CF3HBD. The ligand 6-ACA with an affinity of −4.3 kcal/mol was docked in CF3HBD using AutoDock Vina. **d** Homology model of CF3HBD with ε-caprolactam (CL). The ligand CL with an affinity of −3.7 kcal/mol was docked in CF3HBD using AutoDock Vina. 6-ACA and ε-caprolactam molecules are shown as ball and stick models

control of the T7 promoter and *lacI* regulation. In cells harboring CL-GESS$_{L117F}$ and CF3HBD which were termed ACA-GESS, strong fluorescence was observed in the presence of 5-AVA and 6-ACA (Supplementary Fig. 18). The CF3HBD enzyme can serve as a helper enzyme for substrate detection in the CL-GESS system.

## Discussion

The identification of new enzymes or evolution of enzymes for desired activities is a crucial task for industrial biotechnology. TF-based genetic screening methods have received much attention as enzyme selection systems because they enable the rapid screening of libraries comprising innumerable genetic variants. We previously identified a phenolic compound with GESS, a reporter system that can be used to screen metagenomic or mutation libraries for enzymes[3,25]. Additionally, HTS for the generation of new activity or directed evolution by TF-based biosensor has been reported[26,27]. The TF-based genetic screening method could be useful to generate new activity that would alleviate the current burden on HTS. In this report, we described the design and validation of CL-GESS, a genetically encoded biosensor for the screening and engineering of lactam synthesis enzymes. The engineering of NitR as a TF for higher sensitivity for screening was successful and helpful to screen for a cyclase that has promiscuous activity. We believe that this tool will enable the scientific community to study lactam biosynthesis and further develop an economical bioprocess for lactam production.

NitR is related to the bacterial XylS/AraC family of TFs[17] that are involved in metabolism of carbon sources and have been linked to pathogenesis, with sequence similarity at the carboxyl terminus containing a helix-turn-helix DNA-binding motif[28]. Deletion of NitR resulted in the complete loss of P$_{nitA}$-induced green fluorescence (Supplementary Fig. 4), indicating that NitR activates the transcription of the target gene in the operon. Using engineered NitR in the CL-GESS allowed us to detect target molecules with low background signal levels, which is a desirable feature for HTS systems. Single-cell technologies based on high signal-to-noise ratios have been used to explore cellular mechanisms[29,30]. We applied this strategy to the GESS, although there was a low background signal that was likely due to leaky reporter expression or some other factor. A recent study described the prediction of transcriptional noise based on intracellular physical distance between a regulator (XylS) and the target promoter (Pm)[31]. By random mutagenesis and computational modeling, we identified position 117 in the active pocket of NitR as a critical site for sensitive ligand detection in GESS (Fig. 3d). Wild-type NitR showed a slightly longer distance between ε-caprolactam docked in the active site and the side chain of L117 than did the L117F mutant, suggesting that an aromatic amino acid with a bulky functional group at position 117 favors ε-caprolactam detection.

In this study, we applied CL-GESS to a metagenomic library derived from sea tidal flat sediments to identify enzymes that produce lactam from ω-amino fatty acids. The resulting isolate, CF3HBD is a unique biocatalyst with lactam-synthesizing activity as promiscuous activity that does not require a cofactor or energy source in an in vitro system (Fig. 5a). Along with the CF3HBD, other 3HBD enzymes with high sequence similarity also exhibited the cyclization activity towards 6-ACA (Supplementary Fig. 10). Here, we report an enzyme that can produce ε-caprolactam from 6-ACA identified using metagenomic library screening rather

**Table 1 Data collection and refinement statistics for CF3HBD–NAD$^+$ complex**

| CF3HBD–NAD$^+$ | |
|---|---|
| *Data collection* | |
| Space group | P2$_1$ |
| Cell dimensions | |
| *a, b, c* (Å) | 62.5, 148.3, 62.4 |
| *α, β, γ* (°) | 90, 101.5, 90 |
| Resolution (Å)$^a$ | 50-2.29(2.33-2.29) |
| $R_{sym}$ (%) | 11.8(89.1) |
| $I/\sigma I$ | 28.9(2.4) |
| $R_{meas}$ (%) | 12.8(96.8) |
| $R_{pim}$ (%) | 4.9(37.2) |
| Completeness (%) | 91.8(82.3) |
| Redundancy | 6.8(6.0) |
| *Refinement* | |
| Resolution (Å) | 30.61-2.28 |
| No. reflections | 46,375 |
| $R_{work}/R_{free}$ (%) | 19.69/26.94 |
| No. of atoms | 7708 |
| Protein | 7552 |
| Ligand | 88 |
| Water | 68 |
| Average *B*-factors | 56.84 |
| Wilson *B*-factor (A$^2$) | 43.4 |
| RMS deviations | |
| Bond lengths (Å) | 0.008 |
| Bond angles (°) | 1.072 |
| Ramachandran plot | |
| Favored (%) | 96.1 |
| Allowed (%) | 3.6 |
| Outliers (%) | 0.3 |

$^a$Values in parentheses are for highest-resolution shell

**Table 2 Specific activities of wild-type CF3HBD and mutant enzymes of the active site residues**

| | 3-Hydroxybutyric acid | | 6-Aminocaproic acid | |
|---|---|---|---|---|
| | Specific activity (U/mg) | Relative activity (%) | Specific activity (U/mg) | Relative activity (%) |
| WT | 79.2 ± 2.5 | 100.0 ± 3.2 | 2.1 ± 0.0 | 100.0 ± 0.3 |
| Q91A | –$^a$ | – | – | – |
| Q91E | 90.9 ± 2.3 | 114.9 ± 2.9 | 2.9 ± 0.1 | 139.5 ± 6.4 |
| S139A | – | – | – | – |
| V140A | – | – | 1.7 ± 0.1 | 83.2 ± 4.3 |
| H141A | – | – | – | – |
| H141E | 0.9 ± 0.03 | 1.1 ± 0.04 | 1.2 ± 0.0 | 56.0 ± 0.7 |
| K149A | 7.1 ± 0.4 | 8.9 ± 0.5 | 1.8 ± 0.1 | 88.0 ± 4.8 |
| K149E | 23.5 ± 0.3 | 29.7 ± 0.4 | 1.6 ± 0.0 | 76.5 ± 5.0 |
| Y152A | – | – | 6.5 ± 0.5 | 312.9 ± 23.2 |
| Y152E | 28.5 ± 1.3 | 36.0 ± 1.6 | 2.1 ± 0.7 | 101.5 ± 3.5 |
| W184A | – | – | 1.7 ± 0.2 | 84.2 ± 7.5 |
| W184E | – | – | 1.8 ± 0.1 | 88.0 ± 4.0 |
| W189A | 16.4 ± 0.5 | 20.8 ± 0.7 | 2.2 ± 0.0 | 106.6 ± 1.3 |
| V190A | 77.7 ± 3.6 | 98.2 ± 4.6 | 1.8 ± 0.1 | 86.0 ± 4.7 |
| Q193A | 14.6 ± 0.2 | 18.4 ± 0.3 | 1.0 ± 0.0 | 50.0 ± 1.8 |
| Q193E | 168.1 ± 5.8 | 212.4 ± 7.4 | 2.6 ± 0.2 | 127.5 ± 7.2 |

$^a$No activity at the specified assay conditions

broad substrate specificity[11,12]. In this study, 3-hydroxyacyl-CoA dehydrogenase, acetyl-CoA acetyltransferase, succinyl-CoA:3-ketoacid-CoA transferase, and 3-oxoacid CoA-transferase were also obtained as hits from the metagenomic library screen, suggesting that CL-GESS is sufficiently sensitive to detect low or promiscuous activity. We also developed ACA-GESS, which included CL-GESS$_{L117F}$ and CF3HBD to detect ω-amino fatty acids, such as 5-AVA and 6-ACA, via lactam formation. Thus, we could screen new enzymes using the CL-GESS sensor and apply the results to build another sensor, ACA-GESS. The screening systems presented here are powerful tools for investigating biocatalysts in unknown pathways.

We determined the crystal structure of CF3HBD in complex with NAD$^+$ at a resolution of 2.29 Å to gain insight into its cyclase function. However, the crystal structure of CF3HBD in complex with ε-caprolactam or 6-ACA under various conditions could not obtain diffraction data in spite of various attempts. To identify critical active-site residues and understand the mechanism of ω-amino fatty acid cyclization, we performed mutational analysis based on docking study of 6-ACA to the crystal structure of CF3HBD in complex with NAD$^+$. Alanine substitution mutants of 10 residues located within 4.0 Å of the center of the docked 6-ACA in the crystal structure were studied. Among them, the mutant enzymes of Q91, S139, and H141 completely lost catalytic activities towards 3-hydroxybutyric acid (for dehydrogenation) and 6-ACA (for cyclization), suggesting that both activities share the catalytic site (Table 2).

According to previous reports, enzymatic amide bond formation for lactam synthesis requires the carboxylic group activation. Either ATP or acetyl-CoA acts as an activating molecule to synthesize lactams from ω-amino acids in acyl-CoA ligase[11,12]. Alternatively, CALB lipase catalyzes the amide bond formation for lactam synthesis in organic solvent by transacylation by the active site residues[16]. The mechanism involves a nucleophilic serine in many esterases/lipases of the α/β hydrolase fold superfamily[32,33]. Coincidentally, the S139 residue of CF3HBD was estimated to be in close proximity to the substrate carboxylic group, allowing the enzyme to catalyze the opening of the cyclic amide group in ε-caprolactam, thereby generating 6-ACA as a reaction product, but this activity is very low compared with the cyclization activity (Supplementary Fig. 14). Based on previous reports, the best possible explanation for this finding is the transacylation mechanism but in our best knowledge. Therefore, the cyclization of CF3HBD may follow steps similar to those in the transacylation reaction, as proposed in the reaction scheme in Supplementary Fig. 18. The carboxyl group of the substrate is likely to be activated by the S139 residue acting as a nucleophile, similar to that in other ester-transferase or amide-transferase. This means that S139 and H141 can act together to generate a nucleophile that attacks the carbonyl group to activate the carboxylic group of 6-ACA, while Q91 may form hydrogen bonds with the oxygen of the carbonyl group in 6-ACA (Supplementary Fig. 19). To understand the cyclization activity of CF3HBD in detail, the structure of mutant enzymes with the substrate molecule inside the active site has to be solved. Furthermore, the mechanism of cyclization by CF3HBD needs to be studied in more detail.

In conclusion, we firstly established and engineered a lactam-detecting biosensor (CL-GESS) with high sensitivity and specificity. Using this biosensor, we then screened diverse metagenomes and successfully identified a cyclase that converts 6-ACA to ε-caprolactam. Finally, we determined the X-ray crystal structure of the cyclase to provide insight into its cyclization activity and clarified the active-site residues by mutational study. Using the cyclase found in this study, we next developed a genetically encoded biosensor to sense ω-amino fatty acids (5-ACA or 6-

than through a sequence search. For example, enzymes that convert 6-ACA to ε-caprolactam in the presence of a cofactor or energy source, such as acyl-CoA ligase and β-alanine CoA-transferase, have been identified by sequence searches based on

**Table 3 Kinetic parameters of the wild-type and Y152A mutant enzymes toward 6-aminocaproic acid**

|  | Wild type | Y152A |
|---|---|---|
| $K_m$ (mM) | 3.3 ± 0.3 | 2.9 ± 0.1 |
| $k_{cat}$ (min$^{-1}$ × 10$^{-3}$) | 72.2 ± 2.0 | 208.5 ± 1.6 |
| $k_{cat}/K_m$ (min$^{-1}$ M$^{-1}$) | 21.8 ± 1.8 | 73.1 ± 2.1 |

AVA). Consequently, our CL-GESS could prove a powerful tool for the development of a bioprocess to produce lactams and nylons.

## Methods

**Materials**. All chemical reagents used in this study were purchased from Sigma-Aldrich (St. Louis, MO, USA). Restriction endonucleases, DNA cloning kits, pUC19 plasmid, and *E. coli* DH5α cells were purchased from New England Biolabs (Ipswich, MA, USA). The Diversify PCR Random Mutagenesis kit and Quik-Change II XL Site-Directed Mutagenesis kit were purchased from Clontech (Mountain View, CA, USA) and Agilent Technologies (San Diego, CA, USA), respectively. Plasmid DNA isolation and DNA extraction were conducted using plasmid preparation kits (Promega, Madison, WI, USA). Oligonucleotides (Supplementary Table 5) were synthesized and sequenced by Macrogen (Daejeon, Korea).

**Construction of the CL-GESS plasmid**. We designed an ε-caprolactam detection system consisting of the TR NitR, a NitA promoter, and GFP as a reporter protein. *nitR* and the *nitA* promoter fragments were amplified by PCR from the genomic DNA of *A. faecalis* JM3 KCTC2687 (ATCC8750) using primer pairs 1/2 and 3/4, respectively (Supplementary Table 5). *nitR* was subcloned downstream of the constitutively active promoter J23016 (from http://parts.igem.org), and *nitA* was inserted on the opposite side of *nitR* to avoid transcriptional noise. The *egfp* gene from the plasmid pMGFP[34] was subcloned downstream of the *nitA* promoter. The transcription terminator rrnBT1T2 from plasmid pHCEIIB and tL3 from plasmid pKD46 were inserted at the C-termini of *egfp* and *nitR* by overlap PCR. The CL-GESS plasmid was constructed by ligating the DNA fragment into plasmid pUC19 for *E. coli*, pBBRBB[20] for *P. putida* KT2440, and pSEVA234[35] for *R. eutropha*. We also switched the fluorescent reporter to the *sfgfp*[36] gene to increase the sensitivity of the assay. In order to increase the sensitivity of CL-GESS for ε-caprolactam, we generated various synthetic promoters and RBS for *nitR* transcription by PCR and subcloned these into CL-GESS (Supplementary Table 5). To investigate the NitR-binding site and mechanisms of transcription, truncated promoter fragments concatenated to the *gfp* reporter gene were generated and subcloned into CL-GESS (Supplementary Table 5).

**Identification of the transcription start site in $P_{nitA}$**. Total RNA was extracted from CL-GESS cells cultured with 10 mM ε-caprolactam using the Eastep Super Total RNA Extraction kit (Promega). cDNA was synthesized using the SMARTer RACE cDNA Amplification kit (Clontech). The transcription start site was identified using the Switching Mechanism at the 5′ end of RNA transcript (SMART) RACE cDNA Amplification kit (Clontech) following the manufacturer's protocol. Two reverse gene-specific primers (GSPs) for amplifying *sfgfp* in CL-GESS were designed, including sfgfp-GSP1, which binds the ORF 605 bp downstream of the ATG (5′-tctgctgataatgatctgccagctgca) and was used with the universal primer mix to target the SMART sequence at the 5′ end of the cDNA; and sfgfp-GSP2, located 190 bp downstream of the ATG (5′-aaacactgaacgccataggtca). RACE products were visualized on a 2% agarose gel and were purified using the Wizard SV Gel and PCR Clean-Up System (Promega) according to the manufacturer's protocol. DNA sequencing was performed by Macrogen.

**Mutagenesis of NitR and 3HBD**. A NitR mutant library was constructed by error-prone PCR using a PCR mutagenesis kit with a mutation rate of 2–4 mutations per 1000 base pairs and using CL-GESS as a template. Fragments used to construct the mutant library were prepared by standard PCR and were ligated into the genetic circuit using the Gibson Assembly Master Mix (New England Biolabs). NitR mutants (S133P, L117F, L117A, L117E, L117Q, L117R, L117W, and L117Y) and 3HBD mutants (Q91A, Q91E, S139A, S139E, V140A, V140E, H141A, H141E, K149A, K149E, Y152A, Y152E, Q193A, Q193E, Q193E, W184A, W184E, and V190A) were generated using the QuikChange II XL Site-Directed Mutagenesis kit and a primer pair (Supplementary Tables 6 and 7). Reactions were carried out under conditions specified by the manufacturer. DNA sequencing was performed by Macrogen.

**Mutant library construction and NitR screening**. *E. coli* DH5α cells were transformed by electroporation with ligation mixtures containing mutant genes. Transformants were spread on LB agar plates containing 100 µg/ml ampicillin and were incubated at 37 °C for 16 h. The library (size: $4 \times 10^5$) was stored at −70 °C in storage buffer (1× TY consisting of 8 g tryptone, 5 g yeast extract, and 2.5 g NaCl in 1 l of distilled water) containing 15% (v/v) glycerol until screening by flow cytometry with a FACSAria III instrument (BD Biosciences, Franklin Lakes, NJ, USA). The library ($5.6 \times 10^6$ cells) was inoculated in 2 ml of LB medium containing 100 µM ε-caprolactam and 100 µg/ml ampicillin and incubated at 37 °C for 4 h prior to flow cytometry. A blue laser (488 nm) and bandpass filter (530/30 nm) were used to analyze fluorescence intensity of the mutant library. Approximately 20,000 cells with high fluorescence intensity (top 0.4%) in $5.6 \times 10^6$ cells were collected and were recovered in LB medium at 37 °C for 16 h. In the second round of screening, false-positive cells showing high fluorescence intensity in the absence of ε-caprolactam were removed by sorting non-fluorescent cells. Approximately $1.5 \times 10^5$ (bottom 3%) non-fluorescent cells in $3 \times 10^6$ negatively sorted cells were collected and were grown in LB medium. In the third round, 1000 fluorescent cells with the highest fluorescence in $2 \times 10^6$ (top 0.1%) in the presence of 100 µM ε-caprolactam were sorted and were grown on LB agar, and a single clone exhibiting higher fluorescence intensity than cells expressing wild-type NitR was selected for further examination.

**NitR structure modeling and docking simulations**. Homology modeling of *A. faecalis* NitR was carried out using Discovery Studio 3.1 (Accelrys, San Diego, CA, USA) based on the X-ray structure of ToxT. Homology searches and sequence alignment were conducted using sequence analysis and multiple sequence alignment modules, respectively. Five models were generated based on the alignment of the target sequence with its template using the MODELLER software program[37] by applying the default model-building routine model with fast refinement. The advantages of this procedure are that it allows selection of the best model from among several candidates and that variability among models can be used to evaluate model reliability. Energy minimization was applied using a consistent valence force field and DS CHARMm with the steepest descent and conjugated gradient algorithms. The quality of the models was analyzed with the PROCHECK software[38]. ε-Caprolactam was docked as the ligand in the NitR model using AutoDock Vina[39]. Docking pocket residues were searched using the Pck pocket detection program (http://schwarz.benjamin.free.fr/Work/Pck/home.htm). The lowest energy conformation was selected for further analyses.

**Fluorescence analysis**. *E. coli* DH5α cells harboring CL-GESS and CL-GESS mutants were grown on LB agar containing 100 µg/ml ampicillin and various substrates including ε-caprolactam, L-lysine, and 6-ACA at 37 °C for 16 h. Fluorescent colonies were visualized with an epifluorescence microscope (AZ100-M; Nikon, Tokyo, Japan) equipped with a GFP filter (excitation: 455–485 nm; emission: 500–545 nm). For CL-GESS fluorescence analysis in the liquid phase, single colonies harboring CL-GESS and CL-GESS mutants were inoculated in 2 ml of LB medium with 100 µg/ml ampicillin. After cultivation at 37 °C for 8 h, 1% (v/v) of the seed culture was inoculated in 2 ml of fresh LB with various concentrations of ε-caprolactam (500 µM–30 mM), as well as other chemicals (1 mM) to test ligand specificity. For ACA-GESS fluorescence analysis in the liquid phase, single colonies of *E. coli* EPI300 (DE3) harboring CL-GESS$_{L117F}$ and CF3HBD in pET28a(+) were inoculated in 2 ml of LB medium with 100 µg/ml ampicillin and 25 µg/ml kanamycin. After cultivation at 37 °C for 8 h, 1% (v/v) of the seed culture was inoculated in 2 ml of fresh LB containing ampicillin and kanamycin along with 0.1 mM isopropyl β-D-1-thiogalactopyranoside (IPTG) and various concentrations of ω-amino fatty acids (5-AVA acid and 6-ACA). Fluorescence intensity was measured after 16 h at 37 °C on a multi-label microplate reader (PerkinElmer, Waltham, MA, USA) at excitation and emission wavelengths of 485 and 535 nm, respectively. Cells were sorted on a FACSCalibur instrument (BD Biosciences), and the data were analyzed using the FlowJo software (Tree Star, Ashland, OR, USA).

**Metagenomic library screening by CL-GESS and cyclase**. Metagenomic DNA was isolated from ocean flat tidal sediments on the west coast of Korea (Taean, Korea) using a hydroshear GeneMachine (Genomic Instrumentation Services, Foster City, CA, USA). A metagenomic DNA fosmid library was constructed in *E. coli* EPI300 cells harboring fosmid pCC1FOS using a Copy Control Fosmid Library Production kit (Epicentre, Madison, WI, USA) according to the manufacturer's protocol. For functional screening of cyclase, *E. coli* EPI300 cells harboring the metagenomic library were electrophoretically transformed with the CL-GESS plasmid and were grown on LB agar containing 50 mM 6-ACA, 100 µg/ml ampicillin, and 34 µg/ml chloramphenicol at 37 °C for 14 h. The fluorescence intensity of colonies was evaluated by fluorescence microscopy, and colonies with strong signals were selected and were incubated in LB medium containing 50 mM 6-ACA for 24 h at 37 °C for subsequent screening. Fluorescence intensity was measured with a FACSCalibur instrument (excitation: 488 nm; emission = 530/30 nm) and data were analyzed with FlowJo.

**Cyclase purification and LC–MS analysis**. After screening, the putative cyclase gene (*CF3HBD*) was amplified from the metagenomic hit and cloned into the

pET28a(+) plasmid, yielding plasmid pET28a-cyclase harboring an N-terminal 6× His-tag (Supplementary Table 8). Sufficiently strong expression of CF3HBD was achieved in *E. coli* without codon optimization and the enzyme could be purified at a high yield as shown in Supplementary Fig. 10a. To measure cyclase activity, *E. coli* C2566 cells harboring pET28a(+)-cyclase were grown in LB medium at 37 °C until they reached an optical density at 600 nm of 0.4–0.5. The cells were then induced with 0.1 mM IPTG. The culture was transferred to a 20 °C incubator for 18 h, harvested by centrifugation at 3000 rpm for 10 min, and resuspended in lysis buffer (300 mM KCl, 50 mM KH$_2$PO$_4$, 5 mM imidazole, pH 8.0). Cell extracts were prepared by sonication and purified with the Profinia™ Protein Purification System (Bio-Rad, Hercules, CA, USA), which is configured for automated His-tag affinity chromatography with optional integrated desalting. The final fraction was eluted with 50 mM HEPES buffer (pH 7.5) and was used as a purified enzyme. Protein was quantified by the Bradford method. The purified proteins were confirmed by SDS–PAGE.

**Measurement of activity and kinetic parameters**. The dehydrogenation activity of CF3HBD was determined based on NADH formation from NAD, by measuring the absorbance at 340 nm. The enzymatic reaction was carried out in 50 mM HEPES (pH 7.5) containing 5 mM 3-hydroxybutyric acid, and 10 mM NAD at 40 °C for 10 min. One unit (U) of CF3HBD dehydrogenation activity was defined as the amount of enzyme that catalyzes the reduction of 1 μmol of NAD per min at 40 °C and pH 8.0. For CF3HBD cyclization activity, 1 mM 6-ACA in 50 mM HEPES (pH 7.5) was reacted at 35 °C for 10 min. One unit (U) of CF3HBD cyclization activity was defined as the amount of enzyme required to produce 1 nmol of ε-caprolactam per min at 35 °C and pH 7.5. To examine the effect of pH on the activity of 3HBD, the pH was varied between 6.5 and 8.5 using 50 mM piperazine-*N*,*N*′-bis(2-ethanesulfonic acid) buffer (pH 6.5–7.5) and 50 mM HEPES buffer (pH 7.5–8.5). To investigate the effect of temperature on the enzyme activity, the temperature was varied from 25 to 55 °C. Metal ion-treated enzyme was prepared by adding 0.5 mM metal ion, including Ba$^{2+}$, Ca$^{2+}$, Cu$^{2+}$, Fe$^{2+}$, Mg$^{2+}$, Mn$^{2+}$, Ni$^{2+}$, or Zn$^{2+}$ to the purified CF3HBD enzyme after treatment with 10 mM EDTA, followed by dialysis against 50 mM HEPES (pH 7.5) at 4 °C for 16 h. The enzyme activity was assayed with 1 mM 6-ACA in in 50 mM HEPES (pH 7.5) containing 1 mM metal ion at 35 °C for 10 min. In the kinetics study, various amounts of 6-ACA (0.1–20 mM) were incubated in 50 mM HEPES (pH 7.5) containing purified 3HBD enzyme at 35 °C for 10 min. The reaction was stopped by the addition of HCl at a final concentration of 200 mM. The values of the enzyme kinetic parameters $K_m$ and $k_{cat}$ for substrates were determined by fitting the data to the Michaelis–Menten equation.

**Analytical methods**. ε-Caprolactam and 6-ACA contents in the reaction mixture were determined by LC–MS using an instrument fitted with an Eclipse XDB-C18 column (4.6 × 150 mm; Agilent Technologies, Santa Clara, CA, USA), with a mobile phase consisting of an acetonitrile/1% formic acid (10:90 v/v) gradient from 10% to 90% at 25 °C. The eluent was directed to MS using ESI-positive ion mode with following conditions: fragmentor, 70 V; drying gas flow, 12.0 L/min; drying gas temperature, 350 °C; nebulizer pressure, 35 psig; capillary voltage, 2.5 kV. The scan mode was used and scanned mass range was *m/z* 50–200 kDa. The flow rate was 0.4 ml/min, and retention times of 6-ACA and ε-caprolactam were 3.7 and 8.3 min, respectively. The structure of ε-caprolactam confirmed by 700 MHz $^1$H NMR analysis at the Korea Basic Science Institute (Ochang, Korea) and Infrared (IR) Spectrometer (Bruker EQUINOX 55 FT-IR Spectrometer) at the Korea Research Institute of Chemical Technology (Daejon, Korea), respectively. To confirm the ε-caprolactam from the enzyme reaction solution, ethyl actate extraction method as mentioned below was used. The residue after evaporation was reconstituted with 1/10 volume of water and analyzed by NMR and IR.

To determine the intracellular level of ε-caprolactam, the cultivation at 37 °C for 16 h, 1% (v/v) of the seed culture of CLGESS-harboring *E. coli* DH5α cells were performed in 100 ml of fresh LB with various concentrations of ε-caprolactam (1, 10, and 30 mM). And 50–100 mg wet were washed with 50 ml of saline buffer (1000 times the volume of cells), and then resuspended in 300 μl of 20% trichloroacetic acid to completely degrade the cell membranes. The samples were vortexed for 1 min, incubated on ice for 2 h, and centrifuged at 12,000×*g* at 4 °C for 10 min. The supernatant was collected and mixed with an equal volume of ethyl acetate by vortexing for 10 min. The mixture was centrifuged for 10 min, and the supernatant was transferred to new tube. The ethyl acetate was removed from the extracts by using a rotary evaporator. The residue was reconstituted in an equal volume of PBS buffer and analyzed by LC–MS. To compare extraction efficiencies, we simultaneously extracted a standard of ε-caprolactam.

**Crystallization and structure determination**. Crystals of the CL synthase–NAD$^+$ complex were grown at 20 °C using the hanging-drop vapor diffusion method by mixing 2 μl of protein solution containing 5 mM NAD$^+$ and 2 μl of crystallization buffer with 3.2 M sodium acetate (pH 6.5). Diffraction data were collected at −170 °C using 30% (w/v) glycerol as a cryoprotectant. Crystals belonging to space group P2$_1$, had dimensions *a* = 62.470 Å, *b* = 148.259 Å, and *c* = 62.382 Å, and contained four molecules in an asymmetric unit. Diffraction data from native crystals were

collected using a synchrotron X-ray source at 0.97931 Å on beamline 7A at the Pohang Advanced Light Source (Pohang, South Korea). Datasets were processed using the HKL2000 program. The crystal structure was determined by molecular replacement using the PhaserMR program in the CCP4 software suite[40]. The *P. putida* D-3-hydroxybutyrate dehydrogenase (PDB ID 2Q2Q) sequence was processed using Chainsaw in the CCP4 suite according to the corresponding sequence of the CF3HBD, and was then employed as the search model for CF3HBD[41]. The sequence identity of CF3HBD and PP3HBD is 70.31%. Crystallographic refinement was carried out with PHENIX[41], and model building was performed using COOT[42].

## Data availability

The refined models of CF3HBD have been deposited in the Protein Data Bank (https://www.rcsb.org/structure/5YSS) with PDB code 5YSS. All data that support the findings of this study are included in this article and in Supplementary Information. They are available from the corresponding author upon reasonable request. A reporting summary for this article is available as a Supplementary Information file.

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

## Acknowledgements

This research was supported by grants from C1 Gas Refinery Program funded by the Ministry of Science and ICT (NRF-2018M3D3A1A01055732), the National Research Foundation of Korea (NRF) grant funded by the Korea government (MSIT) (2018R1A2B3004755), the Intelligent Synthetic Biology Center of the Global Frontier Project (2011-0031944), and the Korea Research Institute of Bioscience and Bio-technology Research Initiative Program.

## Author contributions

S.J.Y. designed and performed experiments and wrote the manuscript. S.J.Y. and S.G.L. revised the manuscript and prepared it for submission. M.K., K.K.K., E.R., and performed gene construction and imaging experiments and analyzed data. Y.F. performed analysis of crystal structure and homology modeling. S.H.P. performed site-directed mutagenesis and purified enzymes for determination of active sites. H.L., H.K., D.H.L., and D.M.K. conceived the research and assisted in research design and data interpretation. S.G.L. conceived research, analyzed data, and wrote the manuscript. All authors reviewed the results and approved the manuscript.

## Additional information

**Competing interests:** The authors declare no competing interests.

