## [Peer Review File · Nature Communications]

Reviewers' comments:

Reviewer #1 (Remarks to the Author):

Yeom et al. describe the construction and application of a novel transcription factor-based biosensor for the detection of intracellular ϵ -caprolactam and δ -valerolactam. After initial sensor application and optimization (promoter, RBS and reporter) the sensor was applied to screen a metagenomics library resulting in the identification of a novel cyclase catalysing the conversion of ω -amino fatty acids to lactam. The authors reported the crystal structure of this novel enzyme and also implemented it for the "indirect" sensing of intracellular ω -amino fatty acids via the caprolactam sensor.

Overall, this is a very comprehensive and well-described study. The topic (TF-based biosensors) is of broad interest and the investigate target compounds are highly relevant. -in contrast to several other studies, this one does not stop at the successful sensor design but progresses to the identification of novel enzymes. After consideration of the following comments, I am supportive of publication.

Comments

- 1) l. 92-95 – concerning the tuning of the promoter, RBS, etc. please indicate in the text the used parts from the iGEM repository. From my point of view this is rather a combinatorial analysis and optimization than a "tuning".
- 2) l. 106 – please comment in a bit more detail on the response of the sensor to caprolactam. Which range of concentration. Were all effector molecules applied extracellularly? For caprolactam as the most important effector molecule it would be preferable to relate intracellular effector levels to sensor output.
- 3) Especially the first half of the discussion is rather "review style". Here, the authors should rather focus on the relevance of their sensor system and put this in context with further interesting studies were sensors were applied in mutant or enzyme screenings.
- 4) l. 406-411 – can you comment on how the percentage of positive cells increase from the first to the third screening?
- 5) Reference section – some strange author abbreviations should be corrected. Please put organisms and genes in italics
- 6) Figure 1: a-d is in some details very small – and hardly to read, what is the effector conc used in D? Please include the relevant information in your figure legend! (l. 751: RBS)
- 7) Figure 2: As mentioned above – the measurement of the intracellular caprolactam levels would be important.

Minor comments:

- l. 42 [refs] should likely be filled with some references
- l. 45 "TR-based sensors" – in literature the term "TF-based" (transcription factor rather than regulator) is preferred.
- l. 104 cabable?
- l. 108 PnitA (nitA in italics, also in the following and the figures)
- l. 136 nitR expression (nitR in italics)
- l. 748 PnitA
- l. 755 nitR expression (nitR in italics)

Reviewer #2 (Remarks to the Author):

Yoem et al. have developed a sensitive and specific biosensor for the high-throughput lactam detection, in particular ϵ -caprolactam and related molecules. Next to the engineering of the biosensor, the major part of the manuscript describes the application of the biosensor in identifying novel lactam-synthesizing biocatalysts by screening metagenome libraries with 6-aminocaproic acid as substrate. The authors claim to have found a novel cyclase biocatalyst which directly converts 6-aminocaproic acid to ϵ -caprolactam (an important precursor for nylon production). The authors propose that they have found the first enzymatic amide bond formation without pre-activation of the carboxylic acid (see for example discussion section starting from line 304).

The identification of such an enzyme would be a surprising and outstanding achievement since enzymatic amide bond formation typically depends on carboxylic group activation (see *Angew. Chem. Int. Ed.* 2017, 129, 14690 or *Mol. BioSystem.* 2015, 11, 338 and for ϵ -caprolactam see for example *ACS Synth. Biol.* 2017, 6, 884). In contrast lipases catalyze amide bond formation (as mentioned by the authors) in organic solvent, typically in a transacylation reactions using a leaving group. A direct enzymatic amide bond formation between carboxylic acids and amines (as proposed by the authors) would certainly be a of very high interest for a broad audience ranging from organic chemists to biochemist and synthetic biologists.

Unfortunately, the work on the enzyme characterization of the novel cyclase is not convincing. The authors have solved the structure of the potential cyclase by X-ray crystallography and compared it to related enzyme which catalyzes a completely different reaction (redox reaction, carbonyl reduction). The main issue is that the authors do not show any direct evidence that the identified novel cyclase is catalyzing the direct amide bond formation without preactivation of the carboxylic acid. According to the methods section (line 472) the reactions were carried out with cell lysate which certainly contains ATP and other molecules for carboxylic acid activation. I assume that a big majority of the community would disagree on the proposed reaction scheme based on the current experimental results. To support the author's claim that they have found an enzyme which directly cyclizes omega amino acids (without activation of the carboxylic group), the authors need to present convincing experimental data. Please find further major and minor issues below.

I would fully support publication, if the authors would clearly demonstrate that their cyclase catalyzes a direct amide bond formation without pre-activation of the carboxylic acid.

minor issues

- line 42: References are not given correctly
- line 57: typo 5-aminovaleric acid
- line 60: It is stated that ϵ -caprolactam is mainly chemically synthesized through chemical conversion of omega-amino fatty acids. Can the authors support this by a reference? As far as I know, ϵ -caprolactam is mainly synthesized via a completely different route (Beckmann-rearrangement of the corresponding oxime).
- line 144: text says ϵ -caprolactam was used up to 500 μ M, the figure 2C shows different concentrations. The text reads as 500 μ M is a concentration which shows a strong fluorescence, yet, the figure 2C shows no difference between 0 and 500 μ M lactam concentration which is confusing.
- line 148: The text says that CL-GESS-V6 specifically identifies precursors involved in lactam biosynthesis, which is shown by Fig. S4. In comparison Fig. 2D shows that CL-GESS-V6 can detect various molecules including nitriles, other lactames and ketone analogues. Thus, I do not understand what the statement "specifically identified lactam precursor intermediates" means. Should it mean it only detects ϵ -caprolactam and none of the intermediates in the ϵ -caprolactam biosynthesis (as shown in Fig. S4A)?
- line 263: the given pdb code does not work.

major issues

- line 130: The optimization of CL-GESS is not clear. Replacing GFP with sfGFP generated CL-GESSsfGFP which shows a higher fluorescence response (see Fig 2A). After that, the regulator gene was optimized which yielded CL-GESSsfGFP (same name). It is stated that this optimization also increased the response (line 135), yet this is not shown in Fig 2A. Or does CL-GESSsfGFP reflect an optimization of both, the reporter gene and the regulator gene, then please change the text accordingly.
- line 185: The transcriptional regulator NitR of CL-GESS-V6 was engineered to increase the sensitivity for ϵ -caprolactam. The mutations L117F showed enhanced fluorescence for ϵ -caprolactam (11.4-fold) but also for other molecules such as related lactam's ketones, esters, etc. Did the authors test whether the introduced mutation still showed high selectivity for ϵ -caprolactam compared to the other intermediates of the biosynthetic pathway proposed in Fig. S4A? I understand that this was done with CL-GESS-V6, but was this also done with the final engineered biosensor? I think this is an important control of the final biosensor (not an intermediate biosensor).
- line 256: Why do the authors propose that an unidentified electron density corresponds to the enzyme active site for the cyclase reaction? What is the evidence that the cyclization is really happening in the proposed active site? As mentioned by the authors, the new cyclase enzyme is annotated as hydroxybutyrate dehydrogenase and shows perfect active site conservation with an hydroxybutyrate dehydrogenase of known dehydrogenase function. Why do the authors propose that an amide bond formation is catalyzed by the active site of an enzyme that is proven to catalyze a redox reaction? What is the evidence for this proposal?
- line 324: To me it is not clear why the authors compare and discuss the active site of their cyclase which is proposed to catalyze an amide bond formation with an active site of an alcohol dehydrogenase. I understand that these two enzymes are structurally related, but why do you compare mechanistically important amino acids of the dehydrogenase? What is your mechanistic proposal for the cyclization reaction? Please see very major issue.

Figures & Tables

- There is a typo in Fig 2D. Or should it read Cyclohexaone?
- Table 1 resolution?
- Figures in the downloaded supporting information are not viewable (e.g. Fig. S9)

Reviewer #3 (Remarks to the Author):

The paper by Yeom et al describes the development of a genetic enzyme screening system to identify enzymes that might show activity in the biosynthesis of non-natural lactams, in particular caprolactam. This compound is of interest in biotechnology applications as a precursor for the synthesis of commodities such as nylon-5 and similar polyamides. The screening system is based on the regulatory protein NitR as lactam sensor, which was further engineered for more specific detection of lactams. The authors demonstrate the usefulness of their screening system by screening metagenomes and discover lactam cyclase activity of the 3-hydroxybutyrate dehydrogenase (3HBD) from *Citrobacter freundii*. They also report the crystal structure of this enzyme, which is very similar to the previously determined structure of the homologous 3-hydroxybutyrate dehydrogenase from *Alcaligenes faecalis*. Molecular biosensor/ genetic screening systems are of general interest in biotechnology and a number of such systems have been developed previously, among those also a system to screen for lactam formation (reference 32). There are thus proof-of-principle experiments already published which to some extent lessen the novelty of the paper.

The cyclase activity of 3HBD is unexpected, given the fact that the enzyme belongs to the NAD(P)H dependent short chain dehydrogenase family. Elucidation of the cyclase chemistry would

therefore be of interest. The authors claim in the abstract and the manuscript that they “propose a mechanism for its catalytic action” and “provided insights into its cyclisation activity”. In fact this is not the case. Nowhere in the manuscript are data reported that provide experimental evidence to establish the mechanism of cyclisation, nor is a potential mechanism based on these data discussed in the text. Given the fact that there are known cases of enzymes where NAD⁺/NADH act as a cofactor, but not as redox-co-substrate, it would be interesting to know if this dinucleotide is required for the cyclase activity (is the enzyme completely in the apo form when isolated?). Does NADH inhibit cyclase activity or promote this activity? The authors model the substrate, 6-ACA into the apo-form of the enzyme. Does the substrate overlap with the NAD⁺ binding site? Can it be modelled in the holo-enzyme? Which amino acids in the active site are required for catalytic activity? Experiments to probe if the conserved catalytic triad of this enzyme family is involved in the cyclase activity could also be an important step to elucidate the cyclase mechanism. On page 14, line 325, they mention a conserved lysine residue critical for activity. While this holds for the reaction typical of short chain dehydrogenases it is not known whether this residue is involved in the cyclase reaction. A more detailed analysis of the reaction mechanism would significantly increase the impact of this manuscript.

The in vitro experiments to prove that 3HBD catalyses lactam formation are not described well (page 20). The purification protocol needs clarification (for instance final buffer composition of the enzyme sample; is imidazole removed from the purified samples?) What was the final enzyme concentration in the assay? The authors also need to add LC-MS and NMR of the reaction mixture without enzyme as an important control experiment (Figure 5).

The authors have engineered the sensor protein NitR to be more sensitive to lactams. They have modelled the 3D structure of NitR based on the crystal structure of ToxT from *Vibrio cholera* and used this model to model binding of caprolactam to NitR. In order for the reviewer and the reader to judge the reliability of this model based on a model, they need to provide the amino acid sequence identity between NitA and ToxT. Furthermore, they model the mutants at position L117 of NitR and postulate that in the L117F mutant (see figure 3D) the caprolactam forms pi-pi interactions with the side chain of F117. In the figure it seems however that the distance between the F117 side chain and the carbonyl group of the lactam (the only pi electrons in this molecule) is too large to engage in such an interaction. The authors further write (page 9, line 199) that replacing this residue with a less bulky side chain may increase binding affinity. The opposite however seems true, since the mutants with larger side chains appear more sensitive to caprolactams, and in fact the authors use the L117F mutant for most of their in vivo experiments with the genetic screening system. Please clarify.

The space group is denoted wrongly. There is no space group P21, but the authors probably mean P2₁(subscript)1. Please correct throughout the manuscript.

The structure determination is not described in sufficient detail. They use the homologous structure of the 3HBD from *A. faecalis* as template for the molecular replacement, but it is unclear if this structure was used, a polyalanine model or if the sequence was changed to the corresponding sequence of the *Citrobacter* enzyme. Also the degree of the sequence identity should be given, as well as rmsd values between the two structures.

Minor comments:

On page 7, line 144 it is stated that the caprolactam concentration used for the experiment shown in figure 2C was up to 500 micromolar, whereas in the figure it is up to 30 mM. Please clarify.

Page 10, line 231: it is written that CfHBD was purified as a 34 kD protein, but figure S8A shows a protein band at 28 kDa.

Most of the numbers given in table 1, and in fact in the text page 21, line 488, appear simply be copied from the output of computer programs without much thought about their significance. Examples are for instance cell parameters given to the second and even third decimal, or the percentages of the Ramachandran plot given to the second decimal. I suggest that the authors revise the table and the text by considering the significance of all of these numbers. The Rpim and Rmeas values should be included in the table as well as the Wilson B factor derived from the diffraction data.

Page 3, line 42: insert the proper references.

There are some problems with the references in the reference list: reference 8 is incomplete. Several references only contain initials for the authors (for instance ref 28).

Contrary to the figure legend Figure 6B is not a stereo-view.

Responses to the reviewers' comments

Reviewer #1 (Remarks to the Author):

Yeom et al. describe the construction and application of a novel transcription factor-based biosensor for the detection of intracellular ϵ -caprolactam and δ -valerolactam. After initial sensor application and optimization (promoter, RBS and reporter) the sensor was applied to screen a metagenomics library resulting in the identification of a novel cyclase catalyzing the conversion of ω -amino fatty acids to lactam. The authors reported the crystal structure of this novel enzyme and also implemented it for the “indirect” sensing of intracellular ω -amino fatty acids via the caprolactam sensor. Overall, this is a very comprehensive and well-described study. The topic (TF-based biosensors) is of broad interest and the investigate target compounds are highly relevant. -in contrast to several other studies, this one does not stop at the successful sensor design but progresses to the identification of novel enzymes. After consideration of the following comments, I am supportive of publication.

Comments

1) l. 92-95 – concerning the tuning of the promoter, RBS, etc. please indicate in the text the used parts from the iGem repository. From my point of view this is rather a combinatorial analysis and optimization than a “tuning”.

Response: As suggested by the reviewer, we newly added Supplementary Table 1 listing the promoter and RBS sequences from the iGem repository. In addition, we have rephrased “tuning” as “combinatorial analysis and optimization”.

2) L. 106 – please comment in a bit more detail on the response of the sensor to caprolactam. Which range of concentration. Were all effector molecules applied extracellularly? For caprolactam as the most important effector molecule it would be preferable to relate intracellular effector levels to sensor output.

Response: The relationships of sensor output to caprolactam were investigated in concentration ranges between 0.5~30 mM, while all effector molecules applied extracellularly. As commented by the reviewer, it would be preferable to relate the sensor output with the intracellular effector levels. We tried first to analyze the intracellular caprolactam concentrations based on C^{14} -labeled caprolactam. However, we found that the supply of C^{14} -labeled caprolactam is very limited and expensive (30,000 GBP per 20 μ M C^{14} -labeled caprolactam, Phamaron UK). So, instead we decided to measure the Caprolactam level using a Quadrupole LC-MS (Agilent technologies 6120) used in this study. The experiment were performed as follows: first, the cultivation at 37°C for 16 h, 1% (v/v) of the seed culture of CLGESS-harboring *E. coli* DH5 α cells were performed in 100 ml of fresh LB with various concentrations of ϵ -caprolactam (1,10, and 30 mM). And *E. coli* DH5 α cells (50-100mg) were harvested, and washed in 50 ml of saline buffer (1000 times the volume of cells). Second, treated with 300 μ l of 20% trichloroacetic acid to completely destroy the membrane debris. The samples were vortexed for 1 min, incubated on ice for 2 h, and centrifuged for 10 min at 12,000 $\times g$ at 4°C. The supernatant was extracted with an equal volume of ethyl acetate for 1 min to separate the caprolactam. The mixture was centrifuged

for 1 min, and the supernatant was transferred to new tube. The ethyl acetate was removed from the extracts by using a rotary evaporator. The residual was dissolved in an equal volume of PBS buffer and analyzed by LC-MS as described in the Methods section. As control experiments, we extracted the caprolactam standard solutions simultaneously. The resulting concentration of intracellular caprolactam was estimated to be 0.035–0.27 $\mu\text{mole/mg}$ wet cells when we used 1–30 mM caprolactam extracellularly (Table S2). As the concentration of external caprolactam increased, the intracellular caprolactam concentration increased proportionally (Table S2). In contrast, when the cells were washed twice or three times with 50 ml saline, intracellular caprolactam was not further detectable. These observations suggest that the uptake and release of caprolactam could be equilibrated by passive intracellular transport. Summarizing, we added a supplementary data in Table S2 and mentioned the diffusive result of intracellular caprolactam in the revised manuscript (L150-157).

3) Especially the first half of the discussion is rather “review style”. Here, the authors should rather focus on the relevance of their sensor system and put this in context with further interesting studies where sensors were applied in mutant or enzyme screenings.

Response: As commented by the reviewer, we tried to revise the discussion to show more on the relevance of our sensor system to provide more sensitive and rapid screening systems for enzyme engineering. We added some discussions about the potential of TF-based biosensors to be used as a biosensor for high throughput screenings. The improvement of NitR as the TF to have a higher sensitivity was considered as a critical progress for detecting and breeding a promiscuous activity from a known enzyme. Moreover, combinations of CL-GESS with enzymes such as the cyclase may provide other biosensor systems, as exemplified by the ACA-GESS in the current study. One beneficial point of this system should be its usefulness as a tool to rescue multiple genes for cascade reactions from metagenome, based only on the detection of targeted intermediates. Thus we revised the discussion as follows: “The identification of new enzymes or evolution of enzymes for desired activities is a major goal of industrial biotechnology. TF-based genetic screening methods have received much attention as novel enzyme selection systems because they enable the rapid screening of libraries comprising innumerable genetic variants. We previously identified a phenolic compound with GESS, a reporter system that can be used to screen metagenomic or mutation libraries for enzymes^{3, 24}. Additionally, HTS for the generation of new activity or directed evolution by TF-based biosensor has been reported^{25, 26}. The TF-based genetic screening method could be useful to generate new activity that would alleviate the current burden on high-throughput screening.” In the revised manuscript (L324-331)

4) L. 406-411 – can you comment on how the percentage of positive cells increase from the first to the third screening?

Response: It is difficult to define the positivity rate during the screening rounds because the libraries are the mixture of a wide diversity of variants, not just the defined mixture of positives and negatives. After the screening is completed, we might be able to trace how a final hit was enriched during the screenings. Frankly, we did not trace the final hit of this study but it should have been sorted in 20,000 cells with high fluorescence intensity (top 0.4%) among 5.6×10^6 of 1st round library cells. At the second screening, the false cells maintaining fluorescence regardless of the effector were removed. Approximately 1.5×10^5 (bottom 3%) non-fluorescent cells were sorted and collected among 3×10^6 cells. In the third

round, 1,000 fluorescent cells with the highest fluorescence in 2×10^6 (top 0.1 %) in the presence of 100 μM ϵ -caprolactam were sorted again. We newly added a histogram for the mutant library in the FACS analysis in Supplementary Figure 5 for better understanding.

5) Reference section – some strange author abbreviations should be corrected. Please put organisms and genes in italics

Response: As suggested by the reviewer, the references were corrected.

6) Figure 1: a-d is in some details very small – and hardly to read, what is the effector concentration used in D? Please include the relevant information in your figure legend! (l. 751: RBS)

Response: We appreciate to the comments; the letters were enlarged and the details of relevant information are now included in the revised figure legend.

7) Figure 2: As mentioned above – the measurement of the intracellular caprolactam levels would be important.

Response: As indicated in our response to comment 2, we have measured the intracellular caprolactam levels. After washing of the cells with saline, the intracellular caprolactam levels decreased dramatically, possibly supporting that the uptake and release of caprolactam are equilibrated passively.

Minor comments:

l. 42 [refs] should likely be filled with some references

l. 45 “TR-based sensors” – in literature the term “TF-based” (transcription factor rather than regulator) is preferred.

l. 104 cabable?

l. 108 PnitA (nitA in italics, also in the following and the figures)

l. 136 nitR expression (nitR in italics)

l. 748 PnitA

l. 755 nitR expression (nitR in italics)

Response: Thank you for the detailed comments. All of these concerns were addressed in the revised manuscript.

Reviewer #2 (Remarks to the Author):

Yoem et al. have developed a sensitive and specific biosensor for the high-throughput lactam detection, in particular ϵ -caprolactam and related molecules. Next to the engineering of the biosensor, the major part of the manuscript describes the application of the biosensor in identifying novel lactam-synthesizing biocatalysts by screening metagenome libraries with 6-aminocaproic acid as substrate. The authors claim to have found a novel cyclase biocatalyst which directly converts 6-aminocaproic acid to ϵ -caprolactam (an important precursor for nylon production). The authors propose that they have found the first enzymatic amide bond formation without pre-activation of the carboxylic acid (see for example discussion section starting from line 304).

The identification of such an enzyme would be a surprising and outstanding achievement since enzymatic amide bond formation typically depends on carboxylic group activation (see *Angew. Chem. Int. Ed.* 2017, 129, 14690 or *Mol. BioSystem.* 2015, 11, 338 and for -caprolactam see for example *ACS Synth. Biol.* 2017, 6, 884). In contrast lipases catalyze amide bond formation (as mentioned by the authors) in organic solvent, typically in a transacylation reactions using a leaving group. A direct enzymatic amide bond formation between carboxylic acids and amines (as proposed by the authors) would certainly be a of very high interest for a broad audience ranging from organic chemists to biochemist and synthetic biologists.

Unfortunately, the work on the enzyme characterization of the novel cyclase is not convincing. The authors have solved the structure of the potential cyclase by X-ray crystallography and compared it to related enzyme which catalyzes a completely different reaction (redox reaction, carbonyl reduction). The main issue is that the authors do not show any direct evidence that the identified novel cyclase is catalyzing the direct amide bond formation without preactivation of the carboxylic acid. According to the methods section (line 472) the reactions were carried out with cell lysate which certainly contains ATP and other molecules for carboxylic acid activation. I assume that a big majority of the community would disagree on the proposed reaction scheme based on the current experimental results. To support the author's claim that they have found an enzyme which directly cyclizes omega amino acids (without activation of the carboxylic group), the authors need to present convincing experimental data. Please find further major and minor issues below.

I would fully support publication, if the authors would clearly demonstrate that their cyclase catalyzes a direct amide bond formation without pre-activation of the carboxylic acid.

Response: We thank the reviewer for the constructive comments on the characterization of the cyclase activity. First, we apologize for the confusing description on the preparation of cyclase enzymes in the method section (line 472 in original manuscript). The cyclase has been prepared originally using a Profinia system (Bio-Rad, Hercules, CA, USA), which is an automatic system for purification and desalting of recombinant proteins with a His-affinity tag (Fig. S12). So, we believe the cyclase reactions were carried out with purified enzymes which certainly not contains ATP or other molecules for carboxylic acid activation. The sentence in the method was corrected as follows: "Cell extracts were prepared by sonication and purified with the Profinia™ Protein Purification System (Bio-Rad, Hercules, CA, USA), which is configured for automated His-tag affinity chromatography with optional integrated desalting. The final fraction was eluted with 50 mM HEPES buffer (pH 7.5) and was used as a purified enzyme. Protein was quantified by the Bradford method. The purified proteins were confirmed by SDS-PAGE." (L528-533 in the revised manuscript)

We agree also that the characterization of the cyclase will be critical to prove the claims of this paper. So, we added new data on the comparative characterization of CF3HBD in terms of both the dehydrogenation and cyclization activities. As a result, the purified CF3HBD was found to not require NAD or NADH for cyclization activity toward omega-amino acids (Fig S10), although it belongs to the NAD(P)H-dependent short-chain dehydrogenase family.

Furthermore, to characterize the critical residues involved in both enzyme activities, we conducted a ligand-docking study of the linear form of 6-aminocaproic acid (6-ACA) with

the crystal structure of CF3HBD with NAD⁺ in the active site. From the docking results, 10 residues (Q91, S139, V140, H141, K149, Y152, Q193, W184, V190, and Q193) located within 4.0 Å of the center of the docked substrate were selected as candidate determinant residues for enzyme activity. The selected 10 residues were replaced respectively with alanine or glutamate, and the wild-type and all mutant 3HBD were expressed and purified by HisTrap chromatography as a single band with a molecular mass of approximately 27 kDa in SDS-PAGE (Fig. S11). The mutations at Q91, S139, and H141 completely abolished catalytic activity towards both 3-hydroxybutyric acid and 6-ACA, which suggested that both the dehydrogenation activity and cyclization activity may require these residues (Table 2). Surprisingly, the Y152A mutation removed the dehydrogenation activity but increased the cyclization activity by 2.9 fold. According to a kinetics study, the Y152A mutant showed an approximately 3.4-fold increased catalytic efficiency for 6-ACA, suggesting that small amino acids can be favored for cyclization of 6-ACA in the active site.

Minor issues

- **line 42: References are not given correctly**
- **Line 57: typo 5-aminovaleric acid**

Response: The wrong references and typo were corrected.

1) Line 60: It is stated that -caprolactam is mainly chemically synthesized through chemical conversion of omega-amino fatty acids. Can the authors support this by a reference? As far as I know, -caprolactam is mainly synthesized via a completely different route (Beckmann-rearrangement of the corresponding oxime).

Response: We thank to the reviewer for the comment on caprolactam synthesis. We have revised the phrase about the synthesis of caprolactam as follows; “Caprolactam is most widely used to produce nylon-6 and is mainly produced through Beckmann rearrangement of the cyclohexanone oxime in the presence of fuming sulfuric acid at 90–120°C. Biorenewable routes towards caprolactam from fermentation-derived lysine, muconic acid, adipic acid, and 6-ACA have been discussed¹³. Additionally, the production of 6-ACA by direct fermentation from glucose has also been reported¹⁴.” (L54–59 in the revised manuscript)

2) Line 144: text says -caprolactam was used up to 500µM, the figure 2C shows different concentrations. The text reads as 500µM is a concentration which shows a strong fluorescence, yet, the Figure 2C shows no difference between 0 and 500µM lactam concentration which is confusing.

Response: We apologize for the confusing sentences and have revised the text as follows: “Fluorescence was observed only in the presence of caprolactam (Fig. 2C). The minimal concentration of caprolactam required to activate NitR was 500 µM; moreover, a tight correlation was observed between fluorescence intensity and caprolactam concentration.” (L141–144 in the revised manuscript)

3) Line 148: The text says that CL-GESS-V6 specifically identifies precursors involved in lactam biosynthesis, which is shown by Fig. S4. In comparison Fig. 2D shows that CL-GESS-V6 can detect various molecules including nitriles, other lactams and ketone analogues. Thus, I do not understand what the statement "specifically identified lactam precursor intermediates" means. Should it mean it only detects -caprolactam and none of the intermediates in the caprolactam biosynthesis (as shown in Fig. S4A)?

Response: As presumed by the reviewer, the text was to state that the sensor detects only caprolactam and none of the intermediates in caprolactam biosynthesis were detected. We revised the text appreciatively as follows: “CL-GESSv4 sensed caprolactam, cyclohexanone, *N*-acetylcaprolactam, valerolactam, benzonitrile, and isovaleronitrile, whereas detected none of the intermediates in the caprolactam biosynthesis pathway (Fig. 2D).” (L146–148 in the revised manuscript). In addition, the names of the biosensors were clarified and summarized in Supplementary Fig. 2.

4) Line 263: the given pdb code does not work.

Response: The pdb code was corrected to “2yz7.” The 3HBD from *A. faecalis* (PDB 2yz7) were used one of the model to compare size of active site pocket with CF3HBD to explain cyclization. However, structural alignment between CF3HBD and 3HBD from *A. faecalis* does not important and is rather unclear to explain unexpected cyclization activity of CF3HBD. So we have deleted Fig. S10 in original manuscript and then added new experimental data such as mutational analysis and kinetic study.

Major issues

5) Line 130: The optimization of CL-GESS is not clear. Replacing GFP with sfGFP generated CL-GESSsfGFP which shows a higher fluorescence response (see Fig 2A). After that, the regulator gene was optimized which yielded CL-GESSsfGFP (same name). It is stated that this optimization also increased the response (line 135), yet this is not shown in Fig 2A. Or does CL-GESSsfGFP reflect an optimization of both, the reporter gene and the regulator gene, then please change the text accordingly.

Response: We apologize for the confusing naming of different vectors. All of the names of CL-GESS versions were clarified and are now summarized in Supplementary Fig. 2. The order of construction is as follows: CL-GESSv1→CL-GESSv2→CL-GESSv3→ CL-GESSv4→CL-GESS_{NitR-L117F}.

6) Line 185: The transcriptional regulator NitR of CL-GESS-V6 was engineered to increase the sensitivity for -caprolactam. The mutations L117F showed enhanced fluorescence for -caprolactam (11.4-fold) but also for other molecules such as related lactam’s ketones, esters, etc. Did the authors test whether the introduced mutation still showed high selectivity for -caprolactam compared to the other intermediates of the biosynthetic pathway proposed in Fig. S4A? I understand that this was done with CL-GESS-V6, but was this also done with the final engineered biosensor? I think this is an important control of the final biosensor (not an intermediate biosensor).

Response: Both CL-GESSv4 (previously CL-GESS-V6) and CL-GESS_{NitR-L117F} were investigated for ligand specificity in the revised manuscript (Fig. 2D and Fig. 3C, respectively). The L117F mutation showed high selectivity for caprolactam and did not show fluorescence toward intermediates of the biosynthetic pathway in solid or liquid LB medium (Supplementary Fig. 7). Thus, the final engineered biosensor, CL-GESS_{NitR-L117F} could be used for HTS of new enzymes in the caprolactam biosynthesis pathway. We have revised the relevant sentence as follows; “Various intermediates in the ϵ -caprolactam synthesis pathway, including L-lysine, 5-AVA, and ACA, did not induce sfGFP expression of CL-GESS_{NitR-L117F} in solid or liquid LB medium, suggesting that the highly sensitive CL-GESS_{NitR-L117F} can be used for HTS of new enzymes in the caprolactam biosynthesis pathway (Fig. S7).” (L219–223 in the revised manuscript).

7) line 256: Why do the authors propose that an unidentified electron density corresponds to the enzyme active site for the cyclase reaction? What is the evidence that the cyclization is really happening in the proposed active site? As mentioned by the authors, the new cyclase enzyme is annotated as hydroxybutyrate dehydrogenase and shows perfect active site conservation with a hydroxybutyrate dehydrogenase of known dehydrogenase function. Why do the authors propose that an amide bond formation is catalyzed by the active site of an enzyme that is proven to catalyze a redox reaction? What is the evidence for this proposal?

Response: We agreed that the active site of the new cyclase should not be estimated based only on the ambiguous electron density. Therefore, we conducted additional mutational analyses to determine the important active site residues. As a result, 10 selected residues were separately replaced with alanine or glutamate, and the wild-type and all mutant 3HBD were expressed and purified. Mutation at Q91A, S139A, and H141A completely abolished activity toward 3-hydroxybutyric acid and 6-ACA as shown in Table 2, suggesting that the dehydrogenation activity and cyclization activity of CF3HBD may share the catalytic site. Furthermore, the single mutation of Y152A abolished dehydrogenation activity but resulted in 3-fold enhancement of cyclization activity. According to the kinetics study, the Y152A mutant showed an increase of about 3.4-fold in the catalytic efficiency for 6-ACA (Table 3), suggesting that an alanine residue may be preferred rather than the bulky aromatic amino acid to allow 6-ACA in the active site.

As we mentioned above, the screened CF3HBD belongs phylogenetically to the NAD(P)H-dependent short-chain dehydrogenase family, but it also has cyclization activity as an unexpected, promiscuous activity. The additional cyclization activity of CF3HBD does not require a cofactor or coenzyme in *in-vitro* systems (Fig. S11A and B of revised manuscript).

8) Line 324: To me it is not clear why the authors compare and discuss the active site of their cyclase which is proposed to catalyze an amide bond formation with an active site of an alcohol dehydrogenase. I understand that these two enzymes are structurally related, but why do you compare mechanistically important amino acids of the dehydrogenase? What is your mechanistic proposal for the cyclization reaction? Please see very major issue.

Response: As the reviewer commented, the structural relation of cyclase and alcohol dehydrogenase only would not be sufficient to state the sharing of active site residues between two activities. We tried to add additional evidences by through of the mutational studies of the putative active site residues and comparative characterization of two activities. As a result, we found that both activity were critically dependent on the catalytic residues at positions Q91, S139, and H141. Additionally, the cyclization activity was increased by the Y152A mutant whereas the dehydrogenation activity was completely deleted. Fig. S10 and the related text to compare the active pocket size of CF3HBD and AF3HBD were deleted to clarify the context from the original manuscript.

9) Figures & Tables

- There is a typo in Fig 2D. Or should it read Cyclohexaone?
- Table 1 resolution?
- Figures in the downloaded supporting information are not viewable (e.g. Fig. S9)

Response: : All of these comments were addressed in the revised manuscript.

Reviewer #3 (Remarks to the Author):

The paper by Yeom et al describes the development of a genetic enzyme screening system to identify enzymes that might show activity in the biosynthesis of non-natural lactams, in particular caprolactam. This compound is of interest in biotechnology applications as a precursor for the synthesis of commodities such as nylon-5 and similar polyamides. The screening system is based on the regulatory protein NitR as lactam sensor, which was further engineered for more specific detection of lactams. The authors demonstrate the usefulness of their screening system by screening metagenomes and discover lactam cyclase activity of the 3-hydroxybutyrate dehydrogenase (3HBD) from *Citrobacter freundii*. They also report the crystal structure of this enzyme, which is very similar to the previously determined structure of the homologous 3-hydroxybutyrate dehydrogenase from *Alcaligenes faecalis*. Molecular biosensor/ genetic screening systems are of general interest in biotechnology and a number of such systems have been developed previously, among those also a system to screen for lactam formation (reference 32). There are thus proof-of-principle experiments already published which to some extent lessen the novelty of the paper.

1) The cyclase activity of 3HBD is unexpected, given the fact that the enzyme belongs to the NAD(P)H dependent short chain dehydrogenase family. Elucidation of the cyclase chemistry would therefore be of interest. The authors claim in the abstract and the manuscript that they” propose a mechanism for its catalytic action” and “provided insights into its cyclisation activity”. In fact this is not the case. Nowhere in the manuscript are data reported that provide experimental evidence to establish the mechanism of cyclisation, nor is a potential mechanism based on these data discussed in the text. Given the fact that there are known cases of enzymes where NAD⁺/NADH act as a cofactor, but not as redox-co-substrate, it would be interesting to know if this dinucleotide is required for the cyclase activity (is the enzyme completely in the apo form when isolated?). Does NADH inhibit cyclase activity or promote this activity?

Response: We thank the reviewer for the constructive comments. We conducted new experiments including mutational studies of ten active site residues and characterizations, and thoroughly revised the original manuscript considering the given comments to improve the scope of this paper. As we also responded to the third comment of Reviewer 2, the CF3HBD of this study showed cyclization activity toward omega-amino acids, while it exhibited the highest similarity with the NAD(P)H-dependent short-chain dehydrogenase. However, the presence of NAD or NADH did not have any significant effect on the cyclization activity of purified CF3HBD (Fig. S10). To show the potential residues involved in the cyclase activities, we conducted mutational studies of the active site residues presumed from a ligand-docking with the linear form of 6-ACA. Thus, 10 residues (Q91, S139, V140, H141, K149, Y152, Q193, W184, V190, and Q193) located within 4.0 Å of the center of the docked substrate were selected as candidate determinant residues. The selected 10 residues were separately replaced with alanine or glutamate, and the wild-type and all mutant 3HBD were expressed

and purified by HisTrap chromatography as a single band with molecular mass of approximately 27 kDa in SDS-PAGE (Fig. S11). Alanine substitution at Q91, S139, and H141 completely abolished catalytic activity towards both towards both 3-hydroxybutyric acid and 6-ACA, suggesting that dehydrogenation activity and cyclization activity share the catalytic site (Table 2). Interestingly, the Y152A mutant lost dehydrogenation activity but had 3-fold enhanced cyclization activity. According to a kinetics study, the Y152A mutant showed increase of approximately 3.4-fold in the catalytic efficiency for 6-ACA (Table 3), suggesting that the bulky aromatic ring in the tyrosine residue may be unfavorable to allow 6-ACA in the active site.

2) The authors model the substrate, 6-ACA into the apo-form of the enzyme. Does the substrate overlap with the NAD+ binding site? Can it be modelled in the holo-enzyme? Which amino acids in the active site are required for catalytic activity? Experiments to probe if the conserved catalytic triad of this enzyme family is involved in the cyclase activity could also be an important step to elucidate the cyclase mechanism. On page 14, line 325, they mention a conserved lysine residue critical for activity. While this holds for the reaction typical of short chain dehydrogenases it is not known whether this residue is involved in the cyclase reaction. A more detailed analysis of the reaction mechanism would significantly increase the impact of this manuscript.

Response: We rephrased the descriptions on the modeling studies. In fact, the molecular docking study used the holo-enzyme, not the apo-form of the enzyme. The 6-ACA and NAD molecules are indicated in Figure 6. The Q91A, S139A, and H141A residues that correspond to the conserved catalytic triad of 3HBD enzyme family were all critical for the catalytic activity of the cyclase enzyme. All of these mutations completely abolished activity towards 6-ACA, suggesting that both activities may share the catalytic site. In contrast, the Y152A mutation discriminated between the two activities by deleting the dehydrogenation activity while increasing the cyclization activity by 3.4 fold. The cyclase activity was thus estimated to partly share the active site residues with the dehydrogenase activity, whereas some residues such as Y152 seem to prefer a smaller residue to allow 6-ACA in the active site.

3) The in vitro experiments to prove that 3HBD catalyses lactam formation are not described well (page 20). The purification protocol needs clarification (for instance final buffer composition of the enzyme sample; is imidazole removed from the purified samples?)

Response: As suggested by the reviewer, we added detailed information for the experiment for enzyme characterization and kinetic analysis in the Methods section. Additionally, we described the purification protocol as follows: “Cell extracts were prepared by sonication and purified with the Profinia™ Protein Purification System (Bio-Rad, Hercules, CA, USA), which is configured for automated His-tag affinity chromatography with optional integrated desalting. The final fraction was eluted with 50 mM HEPES buffer (pH 7.5) and was used a purified enzyme.” (L528-533 in the revised manuscript).

4) What was the final enzyme concentration in the assay? The authors also need to add LC-MS and NMR of the reaction mixture without enzyme as an important control

experiment (Figure 5).

Response: We used 0.1 mg/ml purified CF3HBD and 10 mM 6-ACA as substrate at 35°C with 50 mM HEPES buffer (pH 7.5) as had been described in the figure legend. Additionally, we updated data for the scenario without the enzyme (green color) as a control in Fig. 5 A.

5) The authors have engineered the sensor protein NitR to be more sensitive to lactams. They have modelled the 3D structure of NitR based on the crystal structure of ToxT from *Vibrio cholera* and used this model to model binding of caprolactam to NitR. In order for the reviewer and the reader to judge the reliability of this model based on a model, they need to provide the amino acid sequence identity between NitA and ToxT.

Response: We agree that the reliability of a 3D model can be dependent on the amino acid sequence identity with the template. Although there are not many known structures of AraC-type regulators, the NitR model could be generated using ToxT with the closest sequence among the known structures of AraC-type regulators. The overall sequence identity of the two regulators was 18.1% (38.2% similarity) and the identity of the N-terminal ligand binding domain was 23% (43% similarity). The identity levels were newly added to the revised manuscript as follows: To determine the amino acid residues of NitR that are important for the interaction with caprolactam, a homology model of *A. faecalis* NitR was constructed based on the crystal structure of *Vibrio cholerae* O395 ToxT (Protein Data Bank (PDB) entry 3GBG)²¹ as the closest sequence among the known structures of AraC-type regulators. Although the level of sequence identity between NitR and ToxT is relatively low (18.1% identity, 38.2% similarity), the level of sequence identity (23 % identity, 43 % similarity) of the N-terminal domain (substrate binding site) between two regulators can result in useful homology model. (L200-206 in revised manuscript).

6) Furthermore, they model the mutants at position L117 of NitR and postulate that in the L117F mutant (see figure 3D) the caprolactam forms pi-pi interactions with the side chain of F117. In the figure it seems however that the distance between the F117 side chain and the carbonyl group of the lactam (the only pi electrons in this molecule) is too large to engage in such in interaction. The authors further write (page 9, line 199) that replacing this residue with a less bulky side chain may increase binding affinity. The opposite however seems true, since the mutants with larger side chains appear more sensitive to caprolactams, and in fact the authors use the L117F mutant for most of their in vivo experiments with the genetic screening system. Please clarify.

Response: We gratefully accept the comment that the distance between the F117 side chain and the carbonyl group of the lactam can be too large to engage in pi-pi interactions. So, we simplified the sentence and limited the use of hypothetical model only to explain the experimental results (Figure 3D). The largest side chain of L117W appeared more sensitive to caprolactam but it showed a higher background level. Because this high background could be a problem for the screening of large library, we selected L117F for further screening rounds. We have added the following sentence to clarify this: “Although the L117W mutant with larger side chains appeared more sensitive to caprolactam (Fig. S6), it also showed a higher background signal. Therefore, we used L117F mutant for the screening of caprolactam producing enzymes because high background of L117W mutant could be a problem during the screening of large library.” in the revised manuscript (L212–215 in revised manuscript).

7) The space group is denoted wrongly. There is no space group P21, but the authors probably mean P2 (subscript)1. Please correct throughout the manuscript.

Response: We have corrected this.

8) The structure determination is not described in sufficient detail. They use the homologous structure of the 3HBD from *A. faecalis* as template for the molecular replacement, but it is unclear if this structure was used, a polyalanine model or if the sequence was changed to the corresponding sequence of the *Citrobacter* enzyme. Also the degree of the sequence identity should be given, as well as rmsd values between the two structures.

Response: In this study, we used homologous structure of the 3HBD from *P. putida* as template ((PDB ID 2Q2Q) for the molecular replacement as mentioned in method section. On the other hand the 3HBD from *A. faecalis* were used one of the model to compare size of active site pocket with CF3HBD to explain cyclization. However, structural alignment between CF3HBD and 3HBD from *A. faecalis* does not important and is rather unclear to explain unexpected cyclization activity of CF3HBD. So we have deleted Fig. S10 in original manuscript and then added new experimental data such as mutational analysis and kinetic study. Thus we revised method for x-crystal as follows; “The *P. putida* D-3-hydroxybutyrate dehydrogenase (PDB ID 2Q2Q) sequence was processed using Chainsaw in the CCP4 suite according to the corresponding sequence of the CF3HBD, and was then employed as the search model for CF3HBD⁴¹. The sequence identity of CF3HBD and PP3HBD is 70.31%.” in the revised manuscript (L590–594 in revised manuscript).

Minor comments:

9) On page 7, line 144 it is stated that the caprolactam concentration used for the experiment shown in figure 2C was up to 500 micromolar, whereas in the figure it is up to 30 mM. Please clarify.

Response: As commented, the confusing expressions were clarified: “Fluorescence was observed only in the presence of caprolactam (Fig. 2C). The minimal concentration of caprolactam required to activate NitR was 500 μ M; moreover, a tight correlation was observed between fluorescence intensity and caprolactam concentration.” in the revised manuscript (L141–144 in revised manuscript).

10) Page 10, line 231: it is written that CfHBD was purified as a 34 kD protein, but figure S8A shows a protein band at 28 kDa.

Response: The correct size of CF3HBD is 27 kDa. We corrected this in the revised manuscript as pointed out.

11) Most of the numbers given in table 1, and in fact in the text page 21, line 488, appear simply be copied from the output of computer programs without much thought about their significance. Examples are for instance cell parameters given to the second and even third decimal, or the percentages of the Ramachandran plot given to the second decimal. I suggest that the authors revise the table and the text by considering the

significance of all of these numbers. The Rpim and Rmeas values should be included in the table as well as the Wilson B factor derived from the diffraction data.

Response: As suggested by the reviewer, we have added significance values as well as the Rpim and Rmeas values and Wilson B factor in Table 1 in the revised manuscript.

12) Page 3, line 42: insert the proper references.

As suggested by the reviewer, we have added relevant references.

13) There are some problems with the references in the reference list: reference 8 is incomplete. Several references only contain initials for the authors (for instance ref 28).

We have carefully revised the reference list.

14) Contrary to the figure legend Figure 6B is not a stereo-view.

As suggested by the reviewer, we have revised the figure legend of Fig. 6B as follows: “An electron-density map for the active site region”.

Reviewers' comments:

Reviewer #1 (Remarks to the Author):

My comments were appropriately addressed. I am supportive of publication.

Regarding comment 2) Intracellular measurement of caprolactam. We have obtained good results using silicon oil centrifugation as the first step to separate cells from the culture supernatant.

Reviewer #2 (Remarks to the Author):

Many of the previous issues have been addressed in this revised version. Yet, I think there is more data needed to clearly confirm this very unusual enzyme activity. To the best of my knowledge, enzyme-catalyzed amide bond formation in water depends on carboxylic acid pre-activation. To strengthen the claim that this is a promiscuous enzyme function without pre-activation, I would suggest the following experiments, which I think are absolutely necessary as part of the publication.

1) The product of the enzymatic reaction is not fully characterized. So far, the authors present only an LC/MS spectrum and a partial NMR spectrum of a crude product mixture. To confirm the promiscuous enzyme activity, the authors should perform the reaction on larger scale to generate enough product to measure NMR, MS and IR for the isolated and purified caprolactam. This reaction should be performed with the purified enzyme in the absence of any cofactor (such as ATP, NADH, etc.). Since the authors show an HPLC trace with ca. 50% conversion (Fig. 5A), an upscaling experiment (with product isolation) should be straight forward. Please add all the analytical data to the SI.

2) The authors should add data to show the time course of the reaction. This supports that no reaction intermediate arises during the course of the reaction (again, with purified protein and in the absence of any cofactor).

Additional questions/comments (which should make it easier to redo the experiments)

1) Please add the DNA and protein sequences of CF3HBD used in the pET28 plasmid to the SI. Was the gene codon optimized for expression in *E. coli*?

2) How were the units for cyclization activity determined, by HPLC quantification? Please add calibration curves to the SI.

3) Can you please add the data of the kinetic experiments to the SI? How does the Michaelis-Menten plot look like?

Reviewer #3 (Remarks to the Author):

The authors have clarified all my questions concerning the sensor part of the manuscript satisfactorily. However there are still some issues with the unexpected amide bond formation catalyzed by CF3HBD. In my view this is where the real novelty of the paper lies. The mechanism of amide bond formation, and the associated chemical challenge in catalyzing this reaction, is not addressed. As it is this part of the manuscript is descriptive, i.e. it describes the discovery of a novel, unexpected activity but with only limited mechanistic insights.

Other points:

The part of the discussion on the mutants on page 17 is simply a repetition of the results part and can be removed, or better replaced by a more mechanistic discussion. How can a carboxylic acid be activated in the enzyme active site such that amide formation occurs?

The authors have not yet clarified if the enzyme is isolated in its apo- or holo-form. If the latter, the experiments shown in figure S11B are not very meaningful.

I do not quite understand the difference between the panels labelled "standard" and "ε-caprolactam" in figure 5B. Please clarify.

Reviewers' comments:

Reviewer #1 (Remarks to the Author):

My comments were appropriately addressed. I am supportive of publication.

Regarding comment 2) Intracellular measurement of caprolactam. We have obtained good results using silicon oil centrifugation as the first step to separate cells from the culture supernatant.

Response: We appreciate for the positive evaluation. Regarding comment (2), we have known that the use of silicon oil centrifugation will be very useful to measure cell-adsorbed materials in the culture supernatant. However, we also reasoned that it would not be appropriate to apply with caprolactam because it can be extracted substantially into the organic oil layer, as we have experienced with the ethyl acetate extraction in this study (Fig. S9A).

Reviewer #2 (Remarks to the Author):

Many of the previous issues have been addressed in this revised version. Yet, I think there is more data needed to clearly confirm this very unusual enzyme activity. To the best of my knowledge, enzyme-catalyzed amide bond formation in water depends on carboxylic acid pre-activation. To strengthen the claim that this is a promiscuous enzyme function without pre-activation, I would suggest the following experiments, which I think are absolutely necessary as part of the publication.

1) The product of the enzymatic reaction is not fully characterized. So far, the authors present only an LC/MS spectrum and a partial NMR spectrum of a crude product mixture. To confirm the promiscuous enzyme activity, the authors should perform the reaction on larger scale to generate enough product to measure NMR, MS and IR for the isolated and purified caprolactam. This reaction should be performed with the purified enzyme in the absence of any cofactor (such as ATP, NADH, etc.). Since the authors show an HPLC trace with ca. 50% conversion (Fig. 5A), an upscaling experiment (with product isolation) should be straight forward. Please at all the analytical data to the SI.

Response: We appreciate these positive considerations. First, to further characterize the product of the enzymatic reaction, we performed the reaction on a larger scale; purified the product to a single peak in HPLC analysis; and subjected the product to NMR, MS, and IR analyses. The reaction was performed using highly purified enzymes and 10 mM 6-aminocaproic acid without addition of any cofactors. The product was purified to 99% purity by a simple extraction with ethyl acetate (Fig. S9A). The NMR, LC-MS, and IR spectral profiles are displayed in Fig. S9B-D. These results clearly support the synthesis of caprolactam in the absence of any cofactor.

Accordingly, we have revised the manuscript as follows. "Importantly, this reaction was performed using purified enzyme and in the absence of any cofactors such as ATP and NADH. Furthermore, the reaction mixture was extracted with ethyl acetate to purify the reaction product from the other materials. LC-MS analysis revealed that the purity of the obtained caprolactam was greater than 99% (Fig. S9A). Thus, based on the product confirmed by NMR, LC-MS, and IR analyses, we concluded that the promiscuous enzyme catalyzed the cyclization of 6-aminocaproic acid in the absence of any cofactor (Fig. S9 A, C, and D)" (L253-259 in the manuscript)

2) The authors should add data to show the time course of the reaction. This supports that no reaction intermediate arises during the course of the reaction (again, with purified protein and in the absence of any cofactor).

Response: In line with the reviewer's suggestion, we have added new data from experiments conducted to trace the time course of the reaction against various concentrations of purified CF3HBD

(Fig. S13). No reaction intermediate was observed in this time course analysis.

This result has been included in the revised manuscript as follows: “The time course analyses for the production of caprolactam were conducted using 1 mM 6-ACA and CF3HBD. The enzyme concentrations used were 0.1, 0.25, and 0.5 mg/ml and yielded molar conversions of 17.9%, 34.6%, and 47.3%, respectively (Fig. S13). No detectable intermediate was obtained during this reaction” (L276-279 in the revised manuscript)

Additional questions/comments (which should make it easier to redo the experiments)

1) Please add the DNA and protein sequences of CF3HBD used in the pET28 plasmid to the SI. Was the gene codon optimized for expression in *E. coli*?

Response: As suggested, we have added the DNA and protein sequences of *CF3HBD* in the pET28 plasmid to Table S9. *CF3HBD* was amplified by PCR from the obtained metagenomic hit and cloned into the pET28a plasmid. The expression of the gene was high enough without any codon optimization for *E. coli*, and sufficient quantities of the protein were obtained with high purity, as shown in Fig. S10A. We have included the following sentence describing the sequence of *CF3HBD* in the revised manuscript: “The putative cyclase gene (*CF3HBD*) was amplified from the metagenomic hit and cloned into the pET28a plasmid, yielding plasmid pET28a-cyclase harboring an N-terminal 6×His-tag (Table S9). Sufficiently strong expression of CF3HBD was achieved in *E. coli* without codon optimization, and the enzyme could be purified at a high yield, as shown in Fig. S10A” (L548-552 in the revised manuscript).

2) How were the units for cyclization activity determined, by HPLC quantification? Please add calibration curves to the SI.

Response: The unit for cyclization activity was nanomoles of product detected per minute in HPLC quantification. As suggested by the reviewer, the calibration curve for HPLC detection of ϵ -caprolactam has been included in Fig. S9B.

3) Can you please add the data of the kinetic experiments to the SI? How does the Michaelis-Menten plot look like?

Response: As suggested, we have added the kinetic data including the Michaelis–Menten and Lineweaver-Burk plots of 3HBD wild type and the Y152A mutant in Fig. S16. The Y152A mutation seemed more likely to affect the catalytic turnover than the substrate binding affinity. We have revised the manuscript as follows: “Thus, we determined the kinetic parameters using the Michaelis–Menten and Lineweaver-Burk plots of the wild-type and Y152A mutant enzymes toward 6-ACA (Fig. S16).” (L317-319 in the revised manuscript).

Reviewer #3 (Remarks to the Author):

The authors have clarified all my questions concerning the sensor part of the manuscript satisfactorily. However there are still some issues with the unexpected amide bond formation catalyzed by CF3HBD. In my view this is where the real novelty of the paper lies. The mechanism of amide bond formation, and the associated chemical challenge in catalyzing this reaction, is not addressed. As it is this part of the manuscript is descriptive, i.e. it describes the discovery of a novel, unexpected activity but with only limited mechanistic insights.

Response: We appreciate the reviewer’s comment that the sensor part has been clarified. In this revision, we endeavored to address how CF3HBD catalyzes amide bond formation in the absence of cofactors. First, the carboxyl group of the substrate could be activated by the S139 residue acting as

a nucleophile, which is analogous to the catalytic serine residue in catalytic triad-mediated enzymatic catalysis. The Ala-mutations of S139 and H141 reduces the enzyme activity to a non-detectable level, while most other mutations listed in Table 2 allow the enzyme to retain significant levels of its activity. The scheme in Fig. S18 explains that the carboxyl substrate is activated by the interplay of the S139 and H141 and releases spontaneously from the active site following the nucleophilic attack of the substrate amino group that is bended to reach the vicinity of the activated carboxyl substrate. This reaction scheme is rather close to the amide bond formation catalyzed by CALB that catalyzes amide bond formation for lactam synthesis in organic solvent, which is a transacylation reaction via a temporary leaving group created by the action of the catalytic residues, His-Ser. However, it is clearly different from the recent report that the cyclization of ω -amino acids can be detected only after the activation of the carboxylic group by ATP or acetyl-CoA acting as an activating molecule. While the nucleophilic serine in the active site is found in many ester or amide hydrolases, we consistently found that CF3HBD could catalyze the degradation of the cyclic amide group in caprolactam to generate 6-ACA as a reaction product (Fig. S14). This suggests that the cyclization of CF3HBD could follow steps similar to those in the transacylation reaction, as proposed in the reaction scheme in Fig. S18. The X-ray structure obtained in this study suggested that three residues Q91, S139, and H141 are found at active site to explain their critical roles during the cyclization reaction. This means that S139-H141 may act together to generate a nucleophile that attacks the carboxylic group of 6-ACA, while Q91 may form hydrogen bonds with the oxygen of the carbonyl group in 6-ACA.

In the revised manuscript, we have revised the discussion section to include the proposed catalytic mechanism as follows: "According to previous reports, enzymatic amide bond formation for lactam synthesis requires the carboxylic group activation. Either ATP or acetyl-CoA acts as an activating molecule to synthesize lactams from ω -amino acids in acyl-CoA ligase^{10,11}. Alternatively, CALB lipase catalyzes the amide bond formation for lactam synthesis in organic solvent by transacylation by the active site residues¹⁵. The mechanism involves a nucleophilic serine in many esterases/lipases of the α/β hydrolase fold superfamily^{31,32}. Coincidentally, the S139 residue of CF3HBD was estimated to be in close proximity to the substrate carboxylic group, allowing the enzyme to catalyze the opening of the cyclic amide group in ϵ -caprolactam, thereby generating 6-ACA as a reaction product (Fig. S14). Therefore, the cyclization of CF3HBD may follow steps similar to those in the transacylation reaction, as proposed in the reaction scheme in Fig. S18. The carboxyl group of the substrate is likely to be activated by the S139 residue acting as a nucleophile, similar to that in other ester- or amide-transferases. This means that S139 and H141 can act together to generate a nucleophile that attacks the carbonyl group to activate the carboxylic group of 6-ACA, while Q91 may form hydrogen bonds with the oxygen of the carbonyl group in 6-ACA (Fig. S18). To understand the cyclization activity of CF3HBD in detail, the structure of mutant enzymes with the substrate molecule inside the active site have to be solved." (L394-410 in the revised manuscript)

Other points:

The part of the discussion on the mutants on page 17 is simply a repetition of the results part and can be removed, or better replaced by a more mechanistic discussion. How can a carboxylic acid be activated in the enzyme active site such that amide formation occurs? The authors have not yet clarified if the enzyme is isolated in its apo- or holo-form. If the latter, the experiments shown in figure S11B are not very meaningful.

Response: As suggested, we have deleted the redundant material and added mechanistic discussions about the cyclization activity (L394-410 in the revised manuscript). Regarding the apo- or holo-form, all experiments for the cyclase activity were performed using apo-form enzymes, unless the addition of cofactors has been mentioned specifically. Therefore, Fig. S12B demonstrates that the cyclization activity of CF3HBD was not affected by the presence of any cofactors.

I do not quite understand the difference between the panels labelled "standard" and " ϵ -caprolactam" in figure 5B. Please clarify.

Response: We apologize for the confusing terminology. "Standard" and " ϵ -caprolactam" in Fig. 5B refer to the same molecule. To avoid this confusion, we have revised the NMR data in Fig. 5B.

Reviewers' comments:

Reviewer #2 (Remarks to the Author):

The authors have provided all the necessary data to confirm amide bond formation. The proposed reaction mechanism is in analogy to lipase chemistry, however, lipases depend on water elimination to shift the equilibrium. The authors findings are very remarkable and I expect that this manuscript will attract a lot of attention.

Reviewer #3 (Remarks to the Author):

The authors have responded to most of the criticisms/suggestions satisfactorily. There are however a few chemical issues with the mechanistic scheme in supplementary figure S18:

* second and fourth structures from the left: the interaction with the negatively charged oxygen of the intermediates to the side chain amide of Gln91, indicated by the dotted lines, should be to a hydrogen atom, not to the nitrogen of the amide.

* third structure: where is the second hydrogen atom of the amino group of the substrate?

*the authors should reconsider the electron flow indicated by the arrows. They do not make very much chemical sense, in many cases they should be reversed to indicate the correct flow of the electrons in the bond making/bond breaking steps.

Finally, line 623: there is no spacegroup P21, the proper annotation should be P2(subscript)1.

Reviewer #2 (Remarks to the Author):

The authors have provided all the necessary data to confirm amide bond formation. The proposed reaction mechanism is in analogy to lipase chemistry, however, lipases depend on water elimination to shift the equilibrium. The author's findings are very remarkable and I expect that this manuscript will attract a lot of attention.

Response:

Thank you for the encouraging comments. As mentioned by the reviewer, there are no previous reports about a dehydrogenase showing a promiscuous cyclase activity. However, there are several papers about the discovery and engineering of enzymes with novel promiscuous activities (*Angew. Chem. Int. Ed.* 52, 9309–9312 (2013), *Angew. Chem. Int. Ed.* 2016, 55, 4711–4715, *Nature Chemistry*, 2017, 9, 629-634). Many of the activities were extremely weak and hence were not detectable by conventional analyses. Therefore, a critical requirement for detecting these activities is to develop more sensitive analyses. This context suggests that genetically encoded biosensors may be a useful solution for studying promiscuous activity, based on the critical sensitivity of transcription factors. In this paper, a high-affinity CL-sensitive transcription factor could be used to find an unusual cyclization activity in known sequences (Fig. 3 and 4). As commented, the cyclization activity seems unique, particularly because it occurs without the participation of any co-factors for acid activation, differently from other known mechanisms in beta-lactam synthase or in CoA transferase-mediated CL synthesis (*Metab Eng* 2017, 41: 82-91, *ACS Synth Biol* 2016, 5(1): 65-73). Although we cannot demonstrate if the mechanism follows lipase chemistry or not, some of our data provides important clues to estimate the reaction mechanism. First, the enzyme shows amide hydrolysis activity with CL as the substrate, although the activity was extremely low (supplementary Fig. 14). Second, the structural study and mutational analyses revealed that the activity is strongly dependent on the presence of serine and histidine in the active site (Table 2). Third, the substrate amine may participate in the catalysis as it comes to the vicinity of the substrate carboxylic acid. Fourth, 6-aminocaproic acid was detected by the combination of CL-GESS and cyclase activity of 3HBD (supplementary Fig. 17). Considering all experimental data and related literature, we assumed that the mechanism that may start from the presence of nucleophilic serine-histidine network and the substrate amine, and have been described in the manuscript.

In brief, Ser139 and His141 seem to function as a strong nucleophile and an activating base catalyst, respectively, analogous to lipase chemistry. While Ser139 acts as a nucleophilic residue, the substrate amine can now serve as a proton donor to the carboxyl acid group to create a hydroxamate residue in the active site, which could be a better leaving group by yielding a water molecule. The deprotonated amine can now function as a strong nucleophile for the aminolysis of the carbonyl group of the enzyme (supplementary Fig. 18).

Currently, we are very cautious about this putative mechanism because this is the first report on the detection of amide bond formation in aqueous systems without any involvement of acids, solvent, heat, or activating molecules. We understand that more efforts and expertise are required to prove the mechanism, and we have added the following sentences to clarify this point: “To understand the cyclization activity of CF3HBD in detail, the structure of mutant enzymes with the substrate molecule inside the active site has to be solved. Furthermore, the mechanism of cyclization by CF3HBD needs to study in more detail.” (L411-413 in the revised manuscript). We have also carefully added the following sentence in the discussion: “but this activity is very low compared with the cyclization activity (Fig. S14). Based on previous reports, the best possible explanation for this finding is the transacylation mechanism but in our best knowledge.” (L402-405 in the revised manuscript)

In this context, we are very interested in finding a new cyclase with activity homologous to that of lipases. We started this screening by testing commercially available lipases (see the attached Table) in the HEPES buffer (7.5) with 1 mM 6-ACA as the substrate. No commercial lipases showed any detectable cyclization activity in the tested condition. However, we learned from metagenome collections that some natural lipases/esterase family enzymes exhibit marginal cyclase activity, but further investigation of this point is beyond the scope of the present paper.

Table 1. Test of commercially available lipase enzymes for the cyclase activity

Lipases	Lot number	Cyclization activity
Lipase from Candida sp.	L3170 Sigma	N.D
Lipase from Rhizomucor miehei	L4277 Sigma	N.D
Lipase from Thermomyces lanuginosus	L0777 Sigma	N.D
Lipase from porcine pancreas	L3126 Sigma	N.D
Lipase from Aspergillus niger	62301 Sigma	N.D
Lipase from Pseudomonas cepacia	62309 Sigma	N.D
Lipase from Thermus flavus	L3294 Sigma	N.D
Lipase from Thermus thermophilus	L3419 Sigma	N.D
Esterase from porcine liver	E3019 Sigma	N.D

Reviewer #3 (Remarks to the Author):

The authors have responded to most of the criticisms/suggestions satisfactorily.

There are however a few chemical issues with the mechanistic scheme in supplementary figure S18: * second and fourth structures from the left: the interaction with the negatively charged oxygen of the intermediates to the side chain amide of Gln91, indicated by the dotted lines, should be to a hydrogen atom, not to the nitrogen of the amide.* third structure: where is the second hydrogen atom of the amino group of the substrate? *the authors should reconsider the electron flow indicated by the arrows. They do not make very much chemical sense, in many cases they should be reversed to indicate the correct flow of the electrons in

the bond making/bond breaking steps.

Response: We appreciate to the reviewer's kind and detailed comments about the mechanistic scheme of the reaction. We carefully checked the flow of the electrons and corrected all the wrong drawings as suggested (Figure S18 in revised manuscript). Furthermore, we have added sentences in the Discussion as follows: "but this activity is very low compared with the cyclization activity (Fig. S14). Based on previous reports, the best possible explanation for this finding is the transacylation mechanism but in our best knowledge." (L402-405 in the revised manuscript)

Finally, line 623: there is no space group P21, the proper annotation should be P2(subscript)1.

Response: We have revised the manuscript as follows: "Crystals belonged to space group P2₁". (L621 in the revised manuscript)

REVIEWERS' COMMENTS:

Reviewer #2 (Remarks to the Author):

The authors have responded satisfactorily by performing crucial control experiments. Surprisingly, we cannot confirm amide bond formation (or hydrolysis) using the reported conditions, yet, I think this should not further slowdown the publication of this very interesting finding. In my opinion, the authors present all the necessary data to support amide bond formation without pre-activation.

I suggest to change the chemical structure in the Fig. S18. In this proposed mechanism, the structure in the middle has a carbon bearing 5 bonds, which is a bit against the current understanding. The structure can be replaced with a structure of a potential acyl-intermediate which is followed by the attack of the amine.

Responses to Comments

Reviewer #2 (Remarks to the Author):

The authors have responded satisfactorily by performing crucial control experiments. Surprisingly, we cannot confirm amide bond formation (or hydrolysis) using the reported conditions, yet, I think this should not further slowdown the publication of this very interesting finding. In my opinion, the authors present all the necessary data to support amide bond formation without pre-activation. I suggest to change the chemical structure in the Fig. S18. In this proposed mechanism, the structure in the middle has a carbon bearing 5 bonds, which is a bit against the current understanding. The structure can be replaced with a structure of a potential acyl-intermediate which is followed by the attack of the amine.

Response: We appreciate to all the constructive comments by the reviewer. As suggested, we replaced the correspondent chemical structure with a potential acyl-intermediate.